# Neurexophilin4 is a selectively expressed α-neurexin ligand that modulates specific cerebellar synapses and motor functions

Xiangling Meng[1,2], Christopher M McGraw[2,3†], Wei Wang[2,4], Junzhan Jing[1,2], Szu-Ying Yeh[2,3‡], Li Wang[2,4§], Joanna Lopez[2,4], Amanda M Brown[1,2,5], Tao Lin[2,5], Wu Chen[1,2,6], Mingshan Xue[1,2,3,4,6], Roy V Sillitoe[1,2,3,5], Xiaolong Jiang[1,2], Huda Y Zoghbi[1,2,3,4,7]*

[1]Department of Neuroscience, Baylor College of Medicine, Houston, United States; [2]Jan and Dan Duncan Neurological Research Institute at Texas Children's Hospital, Houston, United States; [3]Program in Developmental Biology, Baylor College of Medicine, Houston, United States; [4]Department of Molecular and Human Genetics, Baylor College of Medicine, Houston, United States; [5]Department of Pathology and Immunology, Baylor College of Medicine, Houston, United States; [6]The Cain Foundation Laboratories, Jan and Dan Duncan Neurological Research Institute at Texas Children's Hospital, Houston, United States; [7]Howard Hughes Medical Institute, Baylor College of Medicine, Houston, United States

*For correspondence:
hzoghbi@bcm.edu

Present address: †Department of Neurology, Division of Epilepsy, Massachusetts General Hospital, Boston, United States; ‡Department of Neuroscience, Friedman Brain Institute, Icahn School of Medicine at Mount Sinai, New York, United States; §The Eli and Edythe Broad Center for Regeneration Medicine and Stem Cell Research, University of California, San Francisco, San Francisco, United States

**Abstract** Neurexophilins are secreted neuropeptide-like glycoproteins, and neurexophilin1 and neurexophilin3 are ligands for the presynaptic cell adhesion molecule α-neurexin. Neurexophilins are more selectively expressed in the brain than α-neurexins, however, which led us to ask whether neurexophilins modulate the function of α-neurexin in a context-specific manner. We characterized the expression and function of neurexophilin4 in mice and found it to be expressed in subsets of neurons responsible for feeding, emotion, balance, and movement. Deletion of *Neurexophilin4* caused corresponding impairments, most notably in motor learning and coordination. We demonstrated that neurexophilin4 interacts with α-neurexin and GABA_ARs in the cerebellum. Loss of *Neurexophilin4* impaired cerebellar Golgi-granule inhibitory neurotransmission and synapse number, providing a partial explanation for the motor learning and coordination deficits observed in the *Neurexophilin4* null mice. Our data illustrate how selectively expressed Neurexophilin4, an α-neurexin ligand, regulates specific synapse function and modulates cerebellar motor control.
DOI: https://doi.org/10.7554/eLife.46773.001

## Introduction

Transmission of neural activity requires the signaling of numerous molecules, many of which are specific to certain types of neurons or synapses (*de Wit and Ghosh, 2016*; *Missler et al., 2012*). Among the best-studied and most ubiquitous of these synaptic proteins is the family of neurexins, which are presynaptic cell adhesion molecules consisting of both longer α- and shorter β-isoforms with extensive alternative splicing (*Rowen et al., 2002*). They are proposed to organize synapses throughout the brain (*Südhof, 2008*; *Südhof, 2017*; *Ullrich et al., 1995*) by interacting with trans-synaptic binding partners such as neuroligins (Nlgns) (*Ichtchenko et al., 1995*), neurexophilins (Nxphs) (*Missler et al., 1998*), leucine-rich repeat transmembrane proteins (LRRTMs) (*de Wit et al., 2009*; *Ko et al., 2009*), and Cbln1 (*Matsuda et al., 2010*; *Uemura et al., 2010*). The importance of neurexins to synaptic function is underscored by the fact that mutations in *NRXN1* have been frequently

associated with autism spectrum disorders, intellectual disability, and schizophrenia (*Ching et al., 2010*; *Marshall et al., 2017*; *Feng et al., 2006*; *Gauthier et al., 2011*; *Rujescu et al., 2009*), all neuropsychiatric disorders that have been considered synaptopathies (*Brose et al., 2010*; *Lüscher and Isaac, 2009*). Recent work using mice with conditional deletion of neurexins suggests that the broadly-expressed neurexins perform a synapse-type and circuit specific function (*Anderson et al., 2015*; *Aoto et al., 2015*; *Chen et al., 2017*). Diverse neurexin ligands may play a critical role in mediating neurexin context-specific function, but the detailed mechanisms and their relevence to behavior are largely unknown.

Neuropeptide-like glycoprotein neurexophilin1 and neurexophilin3, both of which have been demonstrated as α-neurexin ligands (*Missler et al., 1998*), are expressed less broadly across the brain than α-neurexin (*Beglopoulos et al., 2005*; *Petrenko et al., 1996*), raising the possibility that Nxphs may modulate the function of α-neurexin trans-synaptic interactions in a context-specific manner. Certain distinct characteristics of Nxphs further support this hypothesis: first, the interaction between Nxphs and α-neurexin is unusually strong, requiring complete denaturation to disrupt them (*Petrenko et al., 1996*). Second, Nxphs are secreted proteins (*Born et al., 2014*), while most other known neurexin ligands are transmembrane proteins. As secreted proteins, Nxphs might serve as modulators for α-neurexin to regulate its binding affinity with specific postsynaptic ligands at individual synapses when multiple ligands are available. Third, both Nxph1 and Nxph3 have been demonstrated to interact with an α-neurexin specific region, the 2$^{nd}$ LNS (Laminin-Neurexin-Sex-hormone-binding globulin) domain (*Missler et al., 1998*), whereas most other neurexin ligands bind to the 6$^{th}$ LNS domain (*Boucard et al., 2005*; *Cheng et al., 2016*; *Ko et al., 2009*; *Siddiqui et al., 2010*; *Zhang et al., 2010*), which is shared by both α- and β-neurexins. Interacting through a binding site different from most other known ligands supports the notion that Nxphs might regulate an α-neurexin complex instead of competing with other ligands for binding sites. In short, the selectively expressed Nxphs could modulate α-neurexin function in specific brain regions to fulfill neurexin's context-specific role, so that loss of this regulation would contribute to the pathogenesis of *NRXN1*-related disorders. Currently, no direct evidence supporting this hypothesis is available, due to lack of tools to study synapse type-specific function of the α-neurexin-Nxphs complex.

Here, we study Nxph4, whose function has not been previously described. We found that Nxph4 is critical for select neural circuits. It interacts with α-neurexin and GABA$_A$Rs in the cerebellum. Deletion of *Nxph4* impairs inhibitory control over cerebellar granule cells, possibly contributing to the motor deficits we observed in the *Nxph4* null mice.

## Results

### *Nxph4* is expressed only in specific brain circuits

Because there is no antibody for endogenous Nxph4 currently available, we generated *Nxph4-βgeo* knock-in mice using targeted embryonic stem cells (ESCs) obtained from the Knock-out Mouse Project (KOMP) repository (*Austin et al., 2004*) to study Nxph4 expression. The construct for this mouse is designed to simultaneously disrupt endogenous *Nxph4* transcript expression while driving expression of a β-galactosidase reporter gene in a promoter-specific manner (*Figure 1—figure supplement 1A and B*), so that β-galactosidase activity indicates *Nxph4* expression. We therefore performed β-galactosidase staining in combination with conventional RNA in situ hybridization to visualize *Nxph4* expression in the brain (*Figure 1—figure supplement 1C*).

*Nxph4* is expressed in subsets of neurons that are interconnected components of several functionally defined brain circuits. For instance, Nxph4 is enriched in the mammillary body circuit, which includes the medial, supramammillary, and lateral mammillary bodies (*Figure 1Ai*), as well as its input sources, the dorsal tegmental nucleus and the presubiculum (*Figure 1Aii*, Aiii, and Aiv) (*Vann and Aggleton, 2004*). Nxph4 is also expressed in two of the three sensory circumventricular organs—the subfornical organ, which is important for controlling fluid balance (*Fry and Ferguson, 2007*; *Johnson and Gross, 1993*; *Zimmerman et al., 2016*), and the area postrema, which is important for energy homeostasis (*Figure 1Bii* and Cii) (*Cottrell and Ferguson, 2004*; *Tan et al., 2016*)—and in nuclei that project to or receive projections from these circumventricular organs. For example, the subfornical organ has reciprocal connections with the medial preoptic nucleus and lateral hypothalamic area (*Fry and Ferguson, 2007*), which are both Nxph4-positive (*Figure 1Bi,Biii*, and Biv). In

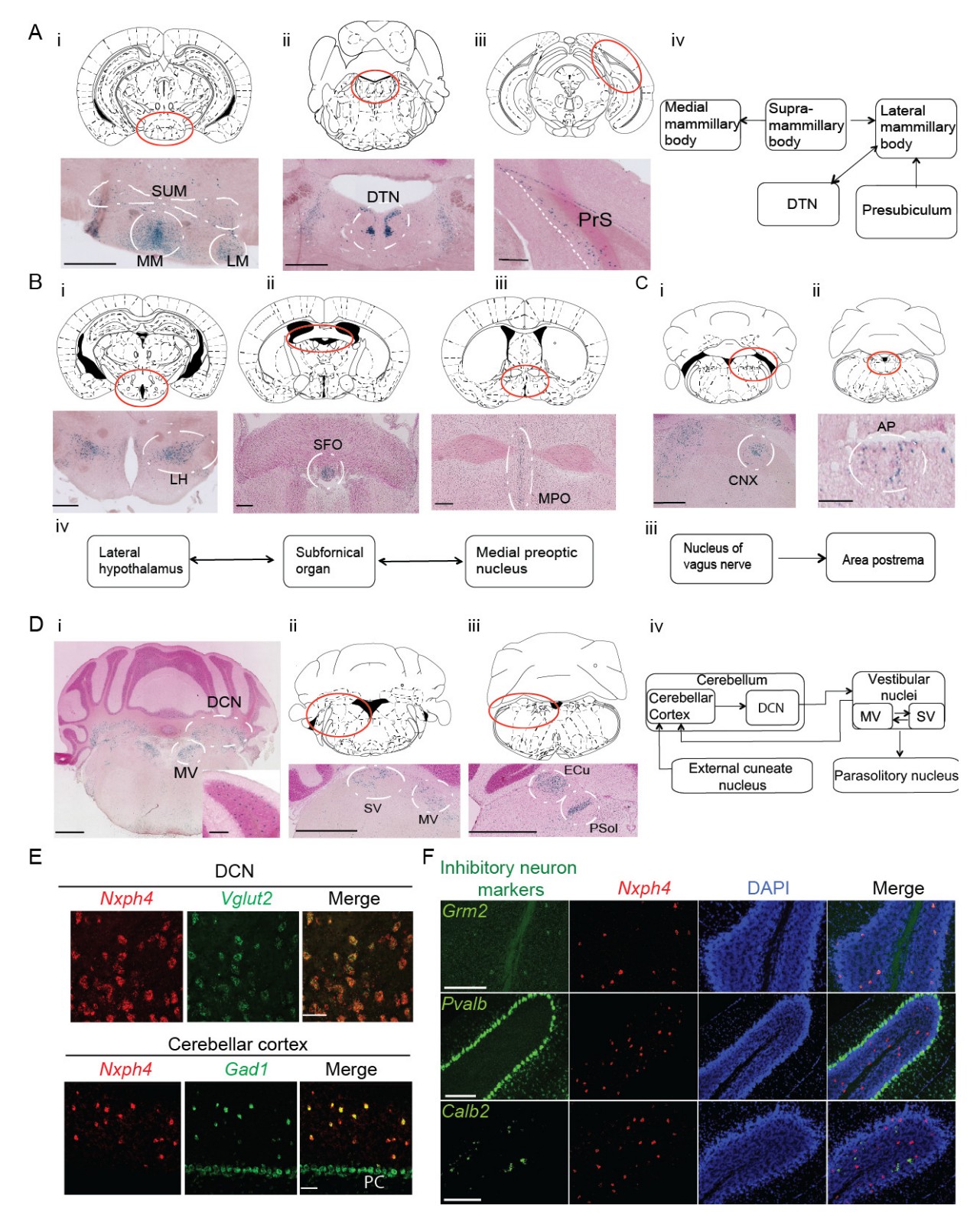

**Figure 1.** Nxph4 expression marks the components of select brain circuits. (A–D) β-galactosidase staining of adult *Nxph4^βgeo/+* mice shows signals in mammillary body-related circuits (A), circumventricular organs (B–C), and cerebellar-vestibular circuits (D). Blue staining represents β-galactosidase activity. Top panels are stereotaxic maps adapted from the Paxinos and Franklin mouse brain atlas, with the red circle indicating the region for the image shown on the bottom panels. Aiv, Biv, Ciii, and Div illustrate the main connections among *Nxph4+* regions in each circuit described. The inset in
*Figure 1 continued on next page*

Figure 1 continued

Di shows the blue staining in the cerebellar cortex granular layer. Scale bars: Ai, 500 µm; Aii, 100 µm; Aiii, 500 µm; Bi 500 µm; Bii, Biii 200 µm; Ci, Cii 100 µm; Di, Dii, Diii, 1 mm; Di inset, 100 µm. SUM, supramammillary body; MM, medial mammillary body; LM, lateral mammillary body; DTN, dorsal tegmental nucleus; PrS, presubiculum; LH, lateral hypothalamus; SFO, subfornical organ; MPO, median preoptic nucleus; CNX, nucleus of vagus nerve; AP, area postrema; DCN, deep cerebellar nuclei; MV, medial vestibular nucleus; SV, superior vestibular nucleus; ECu, external cuneate nucleus; Psol, parasolitary nucleus. (E) Double in situ staining of adult wild type mouse DCN and cerebellar cortex with probes against *Nxph4*, *Vglut2* (an excitatory neuron marker), and *Gad1* (an inhibitory neuron marker). PC: Purkinje cells. Scale bars: 50 µm. (F) Double in situ staining of mouse cerebellum shows that *Nxph4* signals overlap with *Grm2* (a Golgi cell marker) but not *Pvalb* or *Calb2*. Scale bars: 200 µm.
DOI: https://doi.org/10.7554/eLife.46773.002
The following source data and figure supplement are available for figure 1:

**Source data 1.** Brain regions expressing *Nxph4*.
DOI: https://doi.org/10.7554/eLife.46773.004
**Figure supplement 1.** Generation of *Nxph4-βgeo* knock-in mouse and characterization of Nxph4 expression by β-galactosidase staining and in situ hybridization.
DOI: https://doi.org/10.7554/eLife.46773.003

addition, the nucleus of the vagus nerve that projects to area postrema (*Fry and Ferguson, 2007*), also expresses Nxph4 (*Figure 1Ci* and Ciii).

Importantly, *Nxph4* is expressed in several parts of the cerebellar-vestibular circuitry. It is prominent in the medial and superior vestibular nuclei (*Figure 1Di* and Dii), which have reciprocal projections with each other and send information to the cerebellar cortex through mossy fibers (*Figure 1Div*) (*Barmack, 2003*). In the cerebellum, *Nxph4* is expressed in excitatory neurons of the deep cerebellar nuclei and inhibitory neurons in the granular layer, demonstrated by overlapping with the excitatory neuron marker *Slc17a6* (also known as vesicular glutamate transporter 2, referred to as *Vglut2* henceforth), and inhibitory neuron marker *Gad1*, respectively (*Figure 1Di* and E). Specifically, Nxph4 mRNA signal overlaps with Golgi cell marker *Grm2* (*Ohishi et al., 1993*) but not markers of other inhibitory neurons in the cerebellum (*Figure 1F*, *Pvalb*: basket, stellate, and Purkinje cells [*Weyer and Schilling, 2003*]; *Calb2*: Lugaro cells [*Lainé and Axelrad, 2002*]), suggesting that Nxph4 is only expressed in the Golgi cells in the cerebellar cortex. *Nxph4* is visible in the external cuneate nucleus (*Figure 1Diii*), which also projects to the cerebellum through mossy fibers, and the parasolitary nucleus (*Figure 1Diii*), which receives inputs from the vestibular nuclei (Figure Div) (*Barmack, 2003*).

*Nxph4* is in fact expressed in additional locations throughout the brain, including excitatory neurons in the glomerular layer of the main olfactory bulb (*Figure 1—figure supplement 1D*, Ei, and Eii), excitatory neurons in the cerebral cortex layer 6b (*Figure 1—figure supplement 1C and D*), ventral cochlear nucleus (*Figure 1—figure supplement 1Eiii*), pontine nuclei (*Figure 1—figure supplement 1Eiv*), locus coeruleus (*Figure 1Ev*), and other areas as summarized in *Figure 1—source data 1*. The in situ hybridization for *Nxph4* revealed an almost identical expression pattern of *Nxph4* as observed for β-galactosidase staining (*Figure 1—figure supplement 1F* and *Figure 2—figure supplement 1A* left), except for the amygdala (basolateral area and cortical amygdala, *Figure 1—figure supplement 1Fiii*), which did not show strong β-galactosidase activity in the *Nxph4^{βgeo/+}* mouse brain. The specific expression pattern of *Nxph4* in these select neurons suggests that Nxph4 plays a critical role in the physiological function of these circuits, namely motor control, food and energy balance, olfactory function, and emotion.

## *Nxph4* loss leads to reduced weight and anxiety, motor incoordination, and, in male mice, reduced pre-pulse inhibition

To assess the effect of *Nxph4* ablation on brain function, we bred *Nxph4^{βgeo/+}* mice to generate *Nxph4^{βgeo/βgeo}* homozygous knockout mice (KO), which were born at normal Mendelian ratios. There was no specific *Nxph4* RNA signal detected in the KO mice using in situ hybridization with a probe against *Nxph4* (*Figure 2—figure supplement 1A*). We also barely detected Nxph4 mRNA in the KO mice by real time RT-PCR (*Figure 2—figure supplement 1B*). Therefore, we successfully generated a *Nxph4*-null allele.

*Nxph4* KO mice did not show an overt phenotype, but they did gain less weight than controls starting from 5 weeks of age (*Figure 2A*). Their smaller size became more pronounced by the time

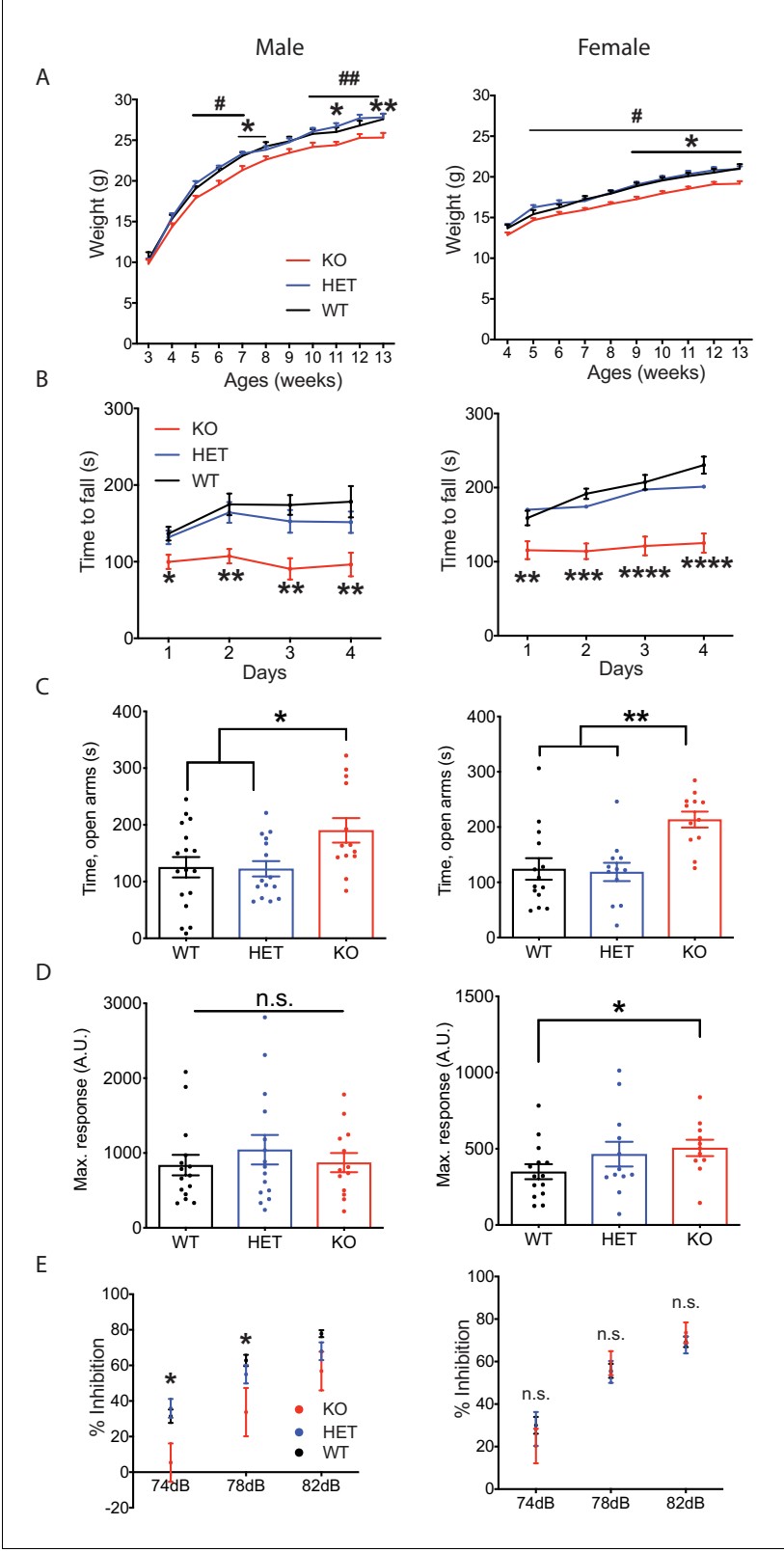

**Figure 2.** *Nxph4* KO mice displayed multiple neurological deficits. (**A**) Plots of weight as a function of age (male n = 13, female n = 16–18; #, difference between HET and KO; * difference between WT and KO). (**B**) Latency to fall from the accelerating rotarod plotted as a function of training days (male n = 10–12, female n = 12). (**C**) Average time spent in the open arms of the elevated plus maze (male n = 13–17, female n = 12–14). (**D**) Mean of response

*Figure 2 continued on next page*

*Figure 2 continued*

to the 120 dB acoustic stimulus (male n = 13–15, female n = 12–14). (E) Pre-pulse inhibition at 74 dB, 78 dB and 82 dB pre-pulses (male n = 13–15, female n = 12–14). Data are presented as mean ± SEM. *, # p<0.05; **p<0.01; ***p<0.001; ****p<0.0001; by one-way or two-way ANOVA.

DOI: https://doi.org/10.7554/eLife.46773.005

The following figure supplement is available for figure 2:

**Figure supplement 1.** *Nxph4* KO mice gained less weight but have normal locomotor and righting reflex functions.

DOI: https://doi.org/10.7554/eLife.46773.006

the mice were 9 months old (**Figure 2—figure supplement 1C and D**). To further characterize the KO mice we then performed a battery of behavioral assays; here we present those relevant to the sites of strong *Nxph4* expression. In the open field test, *Nxph4* KO mice traveled similar distances as controls (**Figure 2—figure supplement 1E and F**). Both male and female *Nxph4* KO mice, however, had difficulty staying on the accelerating rotarod as they only lasted half as long as the heterozygous (*Nxph4$^{\beta geo/+}$*; HET) and wild type (*Nxph4$^{+/+}$*; WT) controls (**Figure 2B**). More strikingly, KO mice failed to improve their performance on this task during four days of training. To examine vestibular function, we performed righting reflex assay on postnatal day 10 mice. Both male and female KO mice took similar time to right themselves after being placed on their back (**Figure 2—figure supplement 1G and H**). Thus, *Nxph4* KO mice had normal locomotor activity and righting reflex but displayed defects in motor coordination and motor learning.

Since Nxph4 is expressed in many regions of the amygdala, we tested the mice for anxiety-like behavior using the elevated plus maze assay. Both male and female KO mice spent significantly more time in the open arms (**Figure 2C**), indicating reduced anxiety. We also performed acoustic startle and pre-pulse inhibition assays to evaluate sensorimotor arousal and gating. Female but not male *Nxph4* KO mice displayed increased response to acoustic stimulus (**Figure 2D**), indicating increased sensorimotor arousal. In contrast, male but not female KO mice showed reduced pre-pulse inhibition when given 74 and 78 dB pre-pulses (**Figure 2E**), suggesting an impairment in sensorimotor gating. It is worth noting that the pre-pulse inhibition defect in the male mice is reminiscent of the greater prevalence of altered sensorimotor gating reported in male patients with schizophrenia and major depression disorder (**Kumari et al., 2004**; **Matsuo et al., 2017**).

## Nxph4 is a secreted glycoprotein

Nxph4 has a similar domain structure as Nxph1, which is a secreted glycoprotein localizing on synapses to interact with α-neurexin (**Figure 3A**) (**Born et al., 2014**; **Missler et al., 1998**). The molecular role of Nxph4, however, remains unknown, and previous studies failed to verify Nxph4 as an α-neurexin ligand (**Missler et al., 1998**). We proposed that Nxph4 is also a secreted glycoprotein that interacts with synaptic proteins. To test our hypothesis, we expressed *Nxph4-3xFLAG-mCherry* in primary cultured cortical neurons using lentivirus infection. We used cortical neurons because it is hard to culture Nxph4-expressing neurons, such as cerebellar Golgi cells. A ~ 62 kDa protein was detected in both the cell lysates and the media, indicating that Nxph4 is secreted to the media (**Figure 3B**). We established that Nxph4 is indeed glycosylated as its molecular weight was reduced with glycosidase treatment or when all of the four asparagine glycosylation sites were mutated to glutamine (Nxph4-4Q-HA, **Figure 3C and D**).

To study Nxph4 subcellular localization and identify Nxph4's interacting partners in vivo, we generated a *Nxph4-3xFLAG* knock-in mouse using the CRISPR-Cas9 system. A triple FLAG tag was inserted into the boundary between the 3$^{rd}$ and 4$^{th}$ domains of Nxph4 (**Figure 3—figure supplement 1A and B**). The anti-FLAG antibody detected a ~ 45 kDa protein in the *Nxph4$^{FLAG/FLAG}$* mice (referred to 'KI mice'), corresponding to the full-length Nxph4 (**Figure 3E**). In the KI but not WT mice, immunofluorescence detected a specific Nxph4-3xFLAG signal only in regions that are positive for Nxph4 (**Figure 3—figure supplement 1C**). Nxph4-3xFLAG is similarly glycosylated (**Figure 3—figure supplement 1D**). The KI mice (with KI allele in absence of wild type allele) showed normal motor coordination and motor learning on the rotarod (**Figure 3—figure supplement 1E**), which indicates that the Nxph4-3xFLAG preserved the function of the wild type protein. In the KI mice, we did not see strong signal of Nxph4 on synapses with immunofluorescence staining, possibly due to

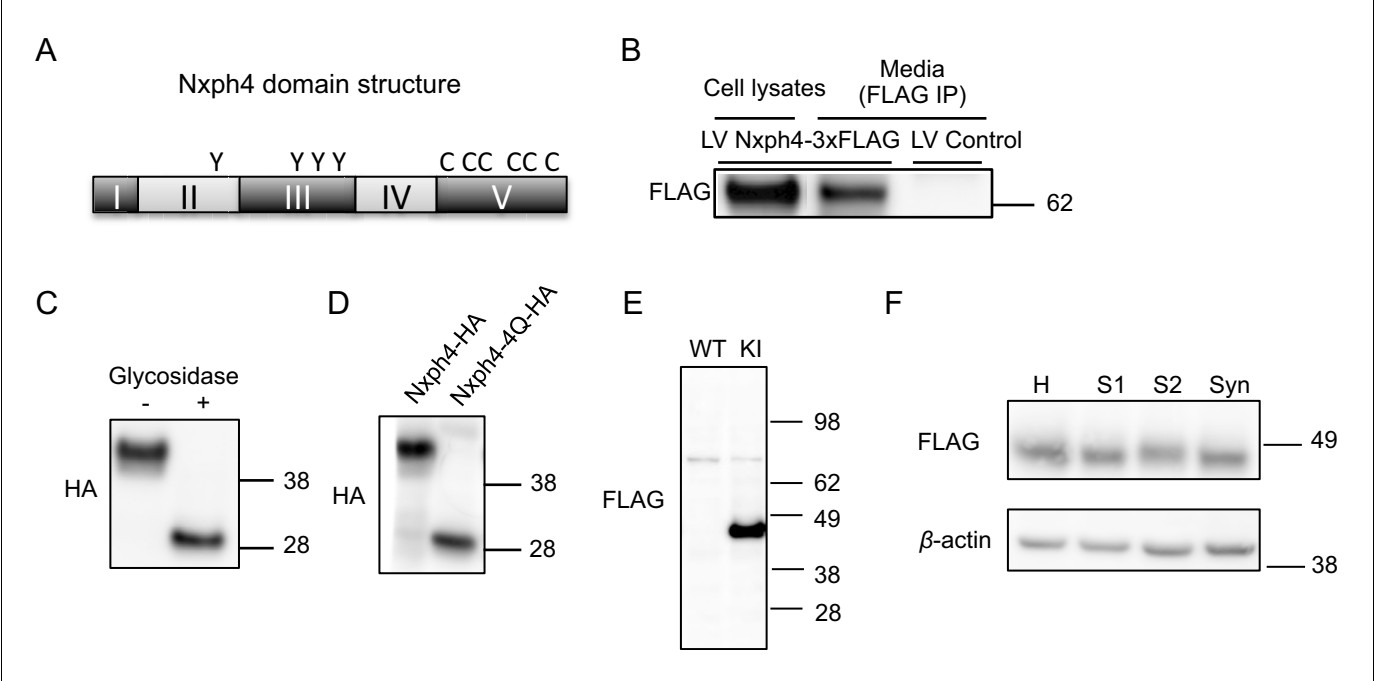

**Figure 3.** Nxph4 is a glycosylated protein that can be detected in the synaptosomes. (**A**) A domain model of Nxph4 (adapted from *Missler and Südhof, 1998*). I: signal peptide; II: a variable domain; III: a conserved domain; IV: a linker region; V: C-terminal domain. Positions of N-glycosylation sequences are marked by letter Y, and the conserved cysteine residues are identified by the letter C. (**B**) Immunoblotting of samples from cultured cortical neurons that are infected with lentivirus expressing Nxph4-3xFLAG or the control lentivirus. Nxph4-3xFLAG was detected in the cell lysates as well as the media. (**C**) Treatment with glycosidase altered the electrophoretic motility of recombinant Nxph4-HA. (**D**) Nxph4-HA-4Q mutant has a smaller molecular mass compared with the wild type recombinant Nxph4-HA. (**E**) Immunoblotting analysis detects Nxph4-3xFLAG expression in the KI mouse synaptosomes. (**F**) Immunoblotting analysis of fractions derived from cerebellar synaptosomal preparation detects Nxph4-3xFLAG in the synaptosomes. β-actin was used as loading control. H: homogenate. S1 and S2 are successive supernatants in the synaptosomal preparation protocol. S2 is also the cytosolic fraction. Syn: synaptosomes.

DOI: https://doi.org/10.7554/eLife.46773.007
The following figure supplements are available for figure 3:

**Figure supplement 1.** Generation and characterization of *Nxph4-3xFLAG* knock-in mice.
DOI: https://doi.org/10.7554/eLife.46773.008
**Figure supplement 2.** Validation of synaptosomes preparation.
DOI: https://doi.org/10.7554/eLife.46773.009

low concentration of Nxph4 after secretion. To probe the presence of Nxph4 at synapses, we prepared subcellular fractionation of the cerebellum. The synaptosomes (Syn) showed enriched synaptic protein PSD-95, while, the cytosolic fraction (S2) showed enriched somatic protein TGFβ−1 (*Figure 3—figure supplement 2A*). Nxph4 is clearly detected in both the cytosolic and synaptosomal fractions, suggesting it is localized at both the synapse and soma (*Figure 3F*). A similar distribution pattern has been reported for other proteins with critical synaptic functions, including Nxph1 (*Reissner et al., 2014*) and RanBP9 (*Palavicini et al., 2013*).

## Nxph4 interacts with α-neurexin in vivo

Because both Nxph1 and Nxph3 are endogenous ligands of α-neurexin (*Ullrich et al., 1995*), we wondered if Nxph4 interacts with α-neurexin. Using the synaptosomes prepared from the KI mice, we were able to pull down Nxph4 by an anti-FLAG antibody. Indeed, we detected α-neurexin in the elution of Nxph4-3xFLAG immunoprecipitation (IP) (*Figure 4A*). To further confirm the α-neurexin-Nxph4 complex in the brain, we performed reciprocal co-IP by pulling down α-neurexin and detected Nxph4-3xFLAG in the elution (*Figure 4B*), further supporting the interaction between α-neurexin and Nxph4. We performed both of the two co-IP experiments in the presence of EDTA, suggesting that Nxph4 interaction with α-neurexin is $Ca^{2+}$ independent. We also precipitated

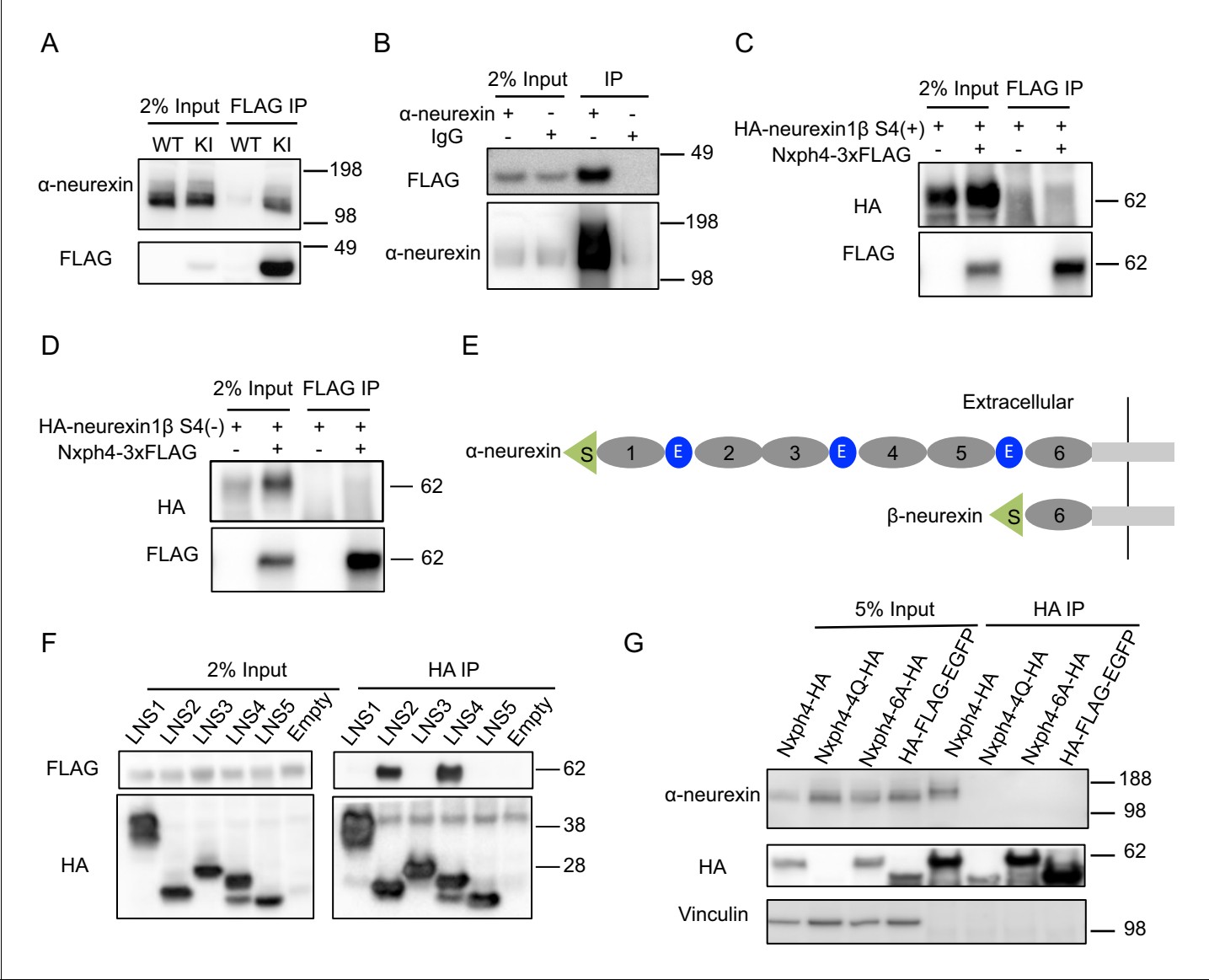

**Figure 4.** Nxph4 forms a complex with α-neurexin in vivo. (A) Synaptosomes (tissue used: olfactory bulb, hypothalamus, midbrain, hindbrain, and the cerebellum) from *Nxph4-3xFLAG* KI or WT (negative control) mice were precipitated with an antibody against FLAG. Bound proteins as well as 2% input were analyzed by immunoblotting with anti-FLAG and anti-α-neurexin antibodies as indicated. (B) Brain lysates from *Nxph4-3xFLAG* KI mice were precipitated with an anti-α-neurexin antibody. Elution and 2% input were analyzed by immunoblotting with anti-α-neurexin and anti-FLAG antibodies. IgG was used as negative control. (C, D) *Nxph4-3xFLAG* and *HA-neurexin1β S4(+)* (with the insertion of splicing site 4, C) or *HA-neurexin1β S4(-)* (without the insertion of splicing site 4, D) were co-expressed in HEK293T cells. Cell lysates were precipitated with an anti-FLAG antibody. Bound proteins were analyzed by immunoblot showing pulling down of Nxph4-3xFLAG but not HA-neurexin1β. Cells transfected with *HA-neurexin1β* alone were used as negative control. (E) Schematic drawing of the extracellular domain structure of α- and β-neurexins. α-neurexin contains 6 LNS domains interspersed by 3 EGF-like repeats. β-neurexin has a single LNS6 domain. S: signal peptide; 1–6: LNS1-6; E: EGF-like domain. (F) Nxph4-3xFLAG was co-expressed with individual α-neurexin specific LNS domains in HEK293T cells. Culture media was precipitated by an anti-FLAG antibody. LNS2 and LNS4 were co-precipitated with Nxph4-3xFLAG. (G) Cultured primary cortical neurons overexpressing wild type or mutant *Nxph4-HA* were subjected to co-IP with an anti-HA antibody. Elution and 5% input were analyzed by immunoblotting with anti-HA, anti-α-neurexin, and anti-vinculin antibodies. Wild type and Nxph4 mutants were fused with mCherry. Nxph4-4Q-HA was not detectable in 5% input and was only detected as a faint band in the IP sample.

DOI: https://doi.org/10.7554/eLife.46773.010

The following figure supplement is available for figure 4:

**Figure supplement 1.** Nxph4 interacts with neurexin1α in vitro and the Nxph4-6A is secreted and glycosylated.

DOI: https://doi.org/10.7554/eLife.46773.011

neurexin1α-Nxph4 complex in HEK293T cells when over-expressing them together in vitro (*Figure 4—figure supplement 1A*). In the same condition, we could not detect β-neurexin interaction with Nxph4 regardless of the presence of splicing site 4 (*Figure 4C and D*), suggesting that Nxph4 is an α-neurexin specific ligand. In addition, we explored which of the α-neurexin specific domains (LNS1-5, *Figure 4E*) might interact with Nxph4 by co-expressing Nxph4 with individual LNS domains in HEK293T cells. We cloned individual LNS domains with an N-terminal signal peptide followed by a HA tag, such that they were secreted into the media after expression in HEK293T cells. In the media, we precipitated Nxph4:LNS2 and Nxph4:LNS4, suggesting that both LNS2 and LNS4 interact with Nxph4 (*Figure 4F*). Thus, our data indicate that Nxph4 is an endogenous α-neurexin ligand.

We further investigated how glycosylation and Nxph4 C-terminal cysteine residues may affect Nxph4 binding affinity to α-neurexin. When we over-expressed an mCherry fused Nxph4-4Q-HA (mutated Nxph4 without glycosylation) in primary cultured cortical neurons through lentivirus infection, we could barely detect its expression in the cell lysates (*Figure 4G*), suggesting that, in neurons, Nxph4 is not stable without glycosylation. On the other hand, Nxph4-6A-HA, with six cysteine residues mutated to alanine, was similarly expressed, glycosylated, and secreted (*Figure 4G*, *Figure 4—figure supplement 1B and C*) as wild-type Nxph4. However, it failed to interact with α-neurexin when over-expressed in the cultured primary cortical neurons (*Figure 4G*), revealing that the cysteine residues are essential for Nxph4 interaction with α-neurexin.

## Nxph4 interacts with GABA$_A$Rs in the cerebellum

In addition to α-neurexin, we wondered whether Nxph4 has any post-synaptic interaction partners. GABA$_A$ receptors are putative candidates because Nxph1 has been shown to recruit post-synaptic GABA$_A$Rs (*Born et al., 2014*). GABA$_A$Rs are pentameric assemblies of subunits, and 19 subunits have been cloned so far (*Sigel and Steinmann, 2012*). Interestingly, GABA$_A$Rα6 is mainly expressed in cerebellar granule cells, the synaptic partner of cerebellar Golgi cells where Nxph4 is expressed (*Eccles et al., 1966*; *Szentagothai, 1965*). To test if the complementary expressed Nxph4 and GABA$_A$Rα6 interact, we prepared cerebellar synaptosomes extracted from *Nxph4-3xFLAG* KI mice and pulled down Nxph4-3xFLAG. GABA$_A$Rα6 was also precipitated in the elution (*Figure 5A*). In the cerebellar granule cells, GABA$_A$Rα1 is another critical subunit in addition to GABA$_A$Rα6 as all GABA$_A$ receptors contain one or both of these two subunits (*Wisden et al., 1996*). In the cerebellum synaptosomal preparation, we also precipitated Nxph4-3xFLAG together with GABA$_A$Rα1 (*Figure 5B*). Moreover, Nxph4 is co-precipitated with the N-terminal extracellular domain of the GABA$_A$ receptors (*Figure 5C and D*), suggesting that it may interact with GABA$_A$ receptors extracellular domain. Taken together, Nxph4 forms a complex with GABA$_A$Rs.

## Loss of *Nxph4* impairs Golgi inhibitory control over granule cells in the cerebellar cortex

As an α-neurexin ligand and an interaction partner of GABA$_A$Rs, Nxph4 may play a role in regulating synaptic function. To determine its physiological role at synapses, we studied synaptic neurotransmission in the *Nxph4* KO mice. We focused on the cerebellar cortex because Nxph4 is solely expressed in the Golgi cells of this region, where they are the main source of inhibitory control over granule cells (*Figure 6A*), the most abundant cell type in the brain. Nxph4 may function at the presynaptic terminal of Golgi cells to regulate Golgi-granule synapses, as other neurexin secreted ligands, such as Cbln1 and C1ql2/3, work at the presynaptic terminals of the cells they are secreted from *Hirai et al. (2005)*; *Matsuda et al. (2016)*. In addition, GABA$_A$Rα1 and α6 are the main α-type GABA$_A$ receptors in cerebellar granule cells (*Wisden et al., 1996*), whose dysfunction has been associated with ataxia in mice (*Chen et al., 1999*; *Payne et al., 2007*). Therefore, Nxph4 may play a critical role in regulating cerebellar Golgi-granule inhibitory synapses through interacting with α-neurexin and GABA$_A$Rs. To test this, we prepared mouse cerebellar slices and performed whole-cell patch clamp recordings on granule cells. We first examined the spontaneous inhibitory postsynaptic current (sIPSC, *Figure 6B*) as a measurement of inhibitory synaptic strength. Comparing with WT mice, we detected a 60% reduction in the frequency of sIPSC in *Nxph4* KO mice (*Figure 6C*). The amplitude was not much altered (*Figure 6D*). This suggests that deletion of *Nxph4* impaired the inhibitory control over granule cells. To further dissect mechanisms underlying the impaired inhibition, we recorded miniature inhibitory postsynaptic current (mIPSC, *Figure 6E*) on the granule cells

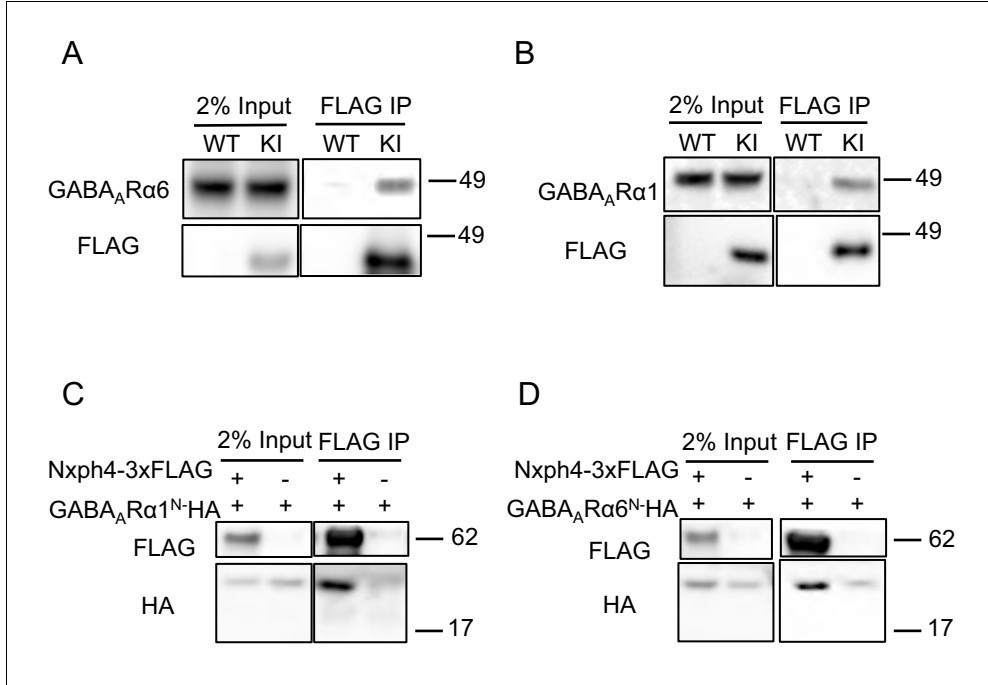

**Figure 5.** Nxph4 interacts with GABA_ARs. (**A,B**) Cerebellar synaptosomes from three *Nxph4-3xFLAG* KI or WT (negative control) mice were precipitated with an antibody against FLAG. Bound proteins as well as 2% input were analyzed by immunoblotting with anti-FLAG and anti-GABA_ARα6 (**A**), or anti-GABA_ARα1 (**B**) antibodies as indicated. (**C**) Nxph4-3xFLAG and the N-terminal extracellular domain of GABA_ARα1 were co-expressed in HEK293T cells. Cell lysates were precipitated with an anti-FLAG antibody and elution was analyzed by immunoblot showing the precipitation of Nxph4-GABA_ARα1$^N$ complex. Cells transfected with *GABA_ARα1$^N$-HA* alone were used as negative control. (**D**) Nxph4-3xFLAG binds to the N-terminal extracellular domain of GABA_ARα6.
DOI: https://doi.org/10.7554/eLife.46773.012

to analyze pre and postsynaptic functions. Consistent with the remarkably reduced sIPSC frequency, mIPSCs frequency in the KO mice was also severely decreased to about half of that in the WT mice (*Figure 6F*). In addition, the amplitude of mIPSCs was also significantly reduced in the *Nxph4* KO mice (*Figure 6G*). There were no significant alterations in mIPSC rise or decay time (*Figure 6—figure supplement 1A and B*). The reduced mIPSC frequency and amplitude imply that, in the *Nxph4* KO mice, both the pre and postsynaptic functions of the inhibitory synapses projected on the granule cells are impaired. Golgi cells provide the main inhibitory inputs on the granule cells. To directly test if loss of Nxph4 impaired Golgi-granule inhibitory synapses and thus resulted in the reduced inhibitory synaptic events detected in KO mice, we used the multi-cell patch recording to compare the connectivity rate of Golgi-granule synapses in WT and KO mice. Golgi cells were identified as large, multipolar cells against densely populated small granule cells in granular layers, and their firing patterns and intrinsic membrane properties were highly distinct from granule cells (*Figure 6I*). We performed whole-cell recording on one Golgi cell first and then recorded its nearby granule cells sequentially to test synaptic connectivity between these two cells while inducing action potentials in Golgi cells (*Figure 6H*). In WT, the probability of finding a connected pair of Golgi-granule cells was ~22% (7 connections out of 32 potential connections tested, *Figure 6I and J*). However, their connectivity rate in KO mice was significantly reduced (1 connection out of 31 potential connections tested,~3%, *Figure 6I and J*). These results directly confirm that loss of Nxph4 impairs Golgi to granule GABAergic synaptic connectivity. In addition to inhibitory transmission, we also examined excitatory inputs projected to granule cells. We recorded evoked excitatory postsynaptic current (eEPSC) from granule cells by stimulating their excitatory inputs, the mossy fibers (*Figure 6—figure supplement 2A–F*). All parameters of eEPSC were comparable across genotype, suggesting that excitatory inputs on the granule cells are not significantly affected after loss of Nxph4. Taken together, Nxph4 may play a critical role in regulating cerebellar Golgi-granule inhibitory synapses, so that deletion of

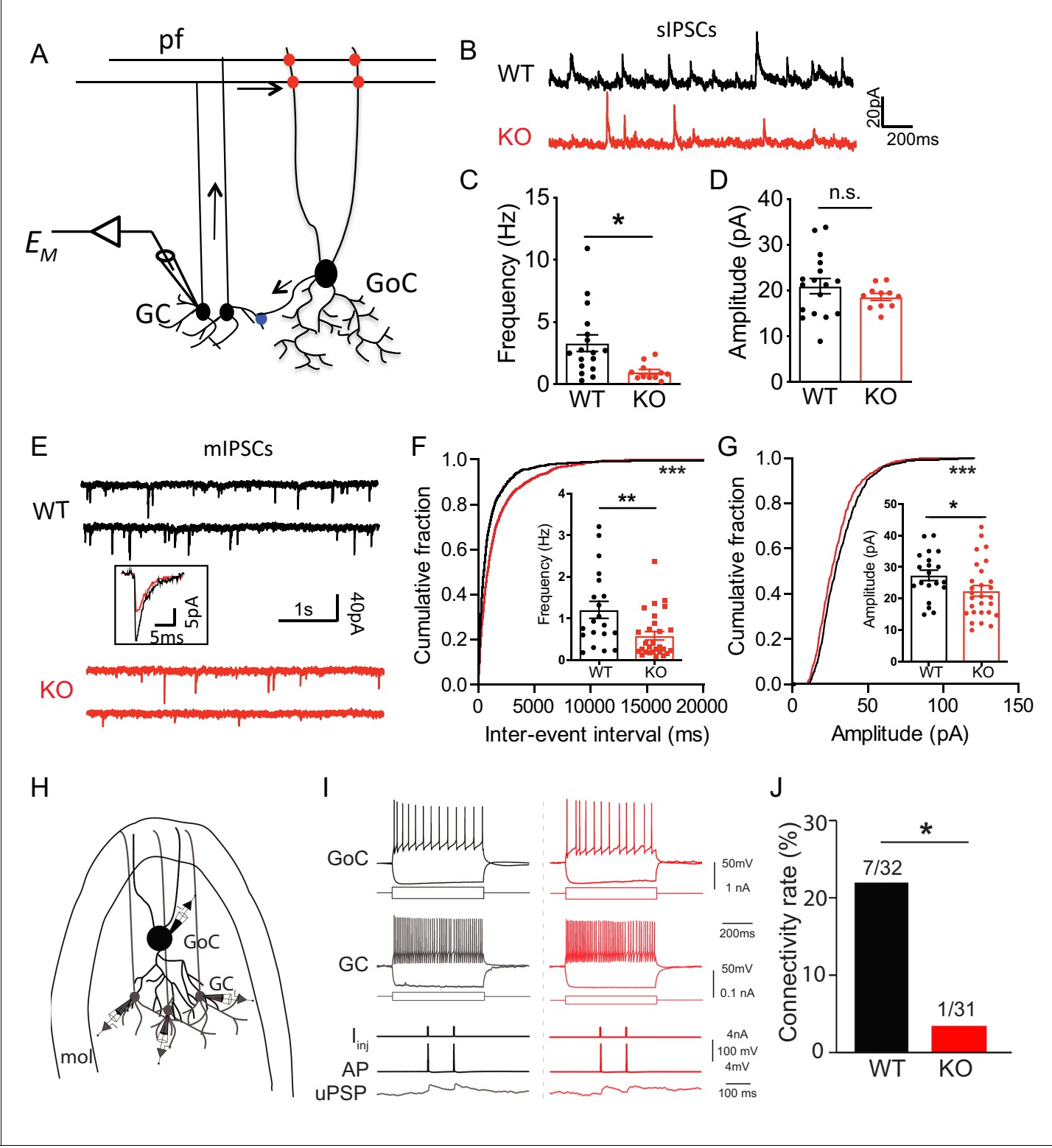

**Figure 6.** Loss of *Nxph4* reduced inhibition onto cerebellar granule cells. (**A**) A simplified diagram illustrating the inhibitory circuit of cerebellar granular layer and single-cell recording experimental design. GC: granule cells; GoC: Golgi Cells; pf: parallel fibers. Arrows indicate the direction of information flow. (**B**) Representative traces of spontaneous IPSC recorded from the WT and KO cerebellar granule cells. (**C–D**) Statistical analysis of sIPSC frequency (C, Mann-Whitney U test) and amplitude (D, t-test) (n = 11–17 cells from 4 WT and 4 KO mice). (**E**) Representative traces of miniature IPSCs recorded from the WT and KO cerebellar granule cells. Insect is the averaged traces. (**F–G**) Statistical analysis of mIPSC frequency (F, Mann-Whitney U test) and

*Figure 6 continued on next page*

*Figure 6 continued*

amplitude (G, t-test). Cumulative probability plots were analyzed by Kolmogorov-Smirnov test (n = 20–28 cells; 6–7 mice). (H) Multi-channel recording configuration. A GoC was first identified and recorded in the granular layer, and then nearby GCs were sequentially recorded to test the connectivity in GOC->GC while inducing action potentials in the GoC. mol: molecular layer. (I) Samples of connected GoC->GC pairs in WT and KO, showing their firing patterns and unitary GABAergic postsynaptic poetntials (uPSP). The recordings were performed in the presence of AMPA and NMDA receptor antagonists to only detect GABAergic synaptic transmission. Given the high-chloride internal solution being used, GABAergic synaptic potentials were depolarized at the resting membrane potentials. $I_{inj}$: injected current; AP: action potential. (J) The connectivity rate (GoC->GC) was significantly lower in KO compared with WT (4 WT and 3 KO mice). Chi-square test. Data are presented as mean ± SEM. n.s., not significant; *p<0.05; ***p<0.001.

DOI: https://doi.org/10.7554/eLife.46773.013

The following figure supplements are available for figure 6:

**Figure supplement 1.** Miniature IPSC decay and rise time were not affected in *Nxph4* KO mice.

DOI: https://doi.org/10.7554/eLife.46773.014

**Figure supplement 2.** Deletion of *Nxph4* KO did not affect mossy fibers-granule cell EPSC.

DOI: https://doi.org/10.7554/eLife.46773.015

**Figure supplement 3.** *Nxph4-3xFLAG* KI mice showed normal mIPSC.

DOI: https://doi.org/10.7554/eLife.46773.016

Nxph4 specifically impaired Golgi inhibitory control over granule cells. There were no significant differences between *Nxph4-3xFLAG* KI and WT mice in the mIPSC recorded on the granule cells (*Figure 6—figure supplement 3A*), further confirming that Nxph4-3xFLAG maintains the function of endogenous Nxph4.

The Golgi-granule inhibitory synapses and the mossy fiber-granule cell excitatory synapses form a typical glomerulus structure in the cerebellar granular layer (*Mapelli et al., 2014*) (*Figure 7—figure supplement 1A*). To test if loss of Nxph4 affects synapse number, we used a combination of pre and postsynaptic markers to label synapses projected on granule cells. Gephyrin and vGAT (vesicular GABA transporter) were used to label inhibitory synapses while PSD-95 and vGlut1 (vesicular glutamate transporter 1) were used for excitatory synapses. A puncta with co-localization of pre and postsynaptic markers was considered as a synapse. In the granular layer, the inhibitory synapse number in the *Nxph4* KO mice was significantly reduced compared with WT (*Figure 7A and B*). However, excitatory synapse number was not significantly altered in the KO mice (*Figure 7C and D*), suggesting that loss of Nxph4 specifically reduced Golgi-granule cell inhibitory synapses. Given the impaired mIPSC amplitude, we also investigated Nxph4 loss-of-function effects on the expression and localization of GABA$_A$Rs. In the cerebellar synaptosomes, we detected similar amount of GABA$_A$Rα1 and α6 in both the KO and WT mice (*Figure 7—figure supplement 1B*). In addition, GABA$_A$Rs showed similar localization and comparable cluster density and size in the two genotypes (*Figure 7—figure supplement 1C and D*). This suggests that Nxph4 might not directly affect GABA$_A$Rs expression or localization. Taken together, deletion of *Nxph4* reduced Golgi-granule cell inhibitory synapse number and impaired the inhibitory neurotransmission onto granule cells.

## Discussion

The mammalian central nervous system comprises numerous neuronal types with distinct electrophysiological properties. The advantage of such complex circuitry is to adapt rapidly and reversibly to neuronal activity through any component in the circuit. Failing in this process would generate profound disturbance in the circuits, including altered excitatory and inhibitory balance, which has been proposed to be a key etiology of neuropsychiatric disorders, including autism spectrum disorders and intellectual disabilities (*Nelson and Valakh, 2015*; *Rubenstein and Merzenich, 2003*). Current evidence suggests that cell type-specific proteins form the molecular basis for fine regulation of diverse synapses to fulfill circuit function (*de Wit and Ghosh, 2016*; *Margeta and Shen, 2010*). Here, we show that Nxph4 is expressed in select neurons of several neural circuits. Focusing on its expression in the cerebellar Golgi cells, we performed studies at the molecular, synaptic, and behavioral levels and discovered that Nxph4 interacts with both α-neurexin and GABA$_A$Rs and regulates Golgi-granule cell inhibitory synapse function, which is critical for motor coordination and motor learning.

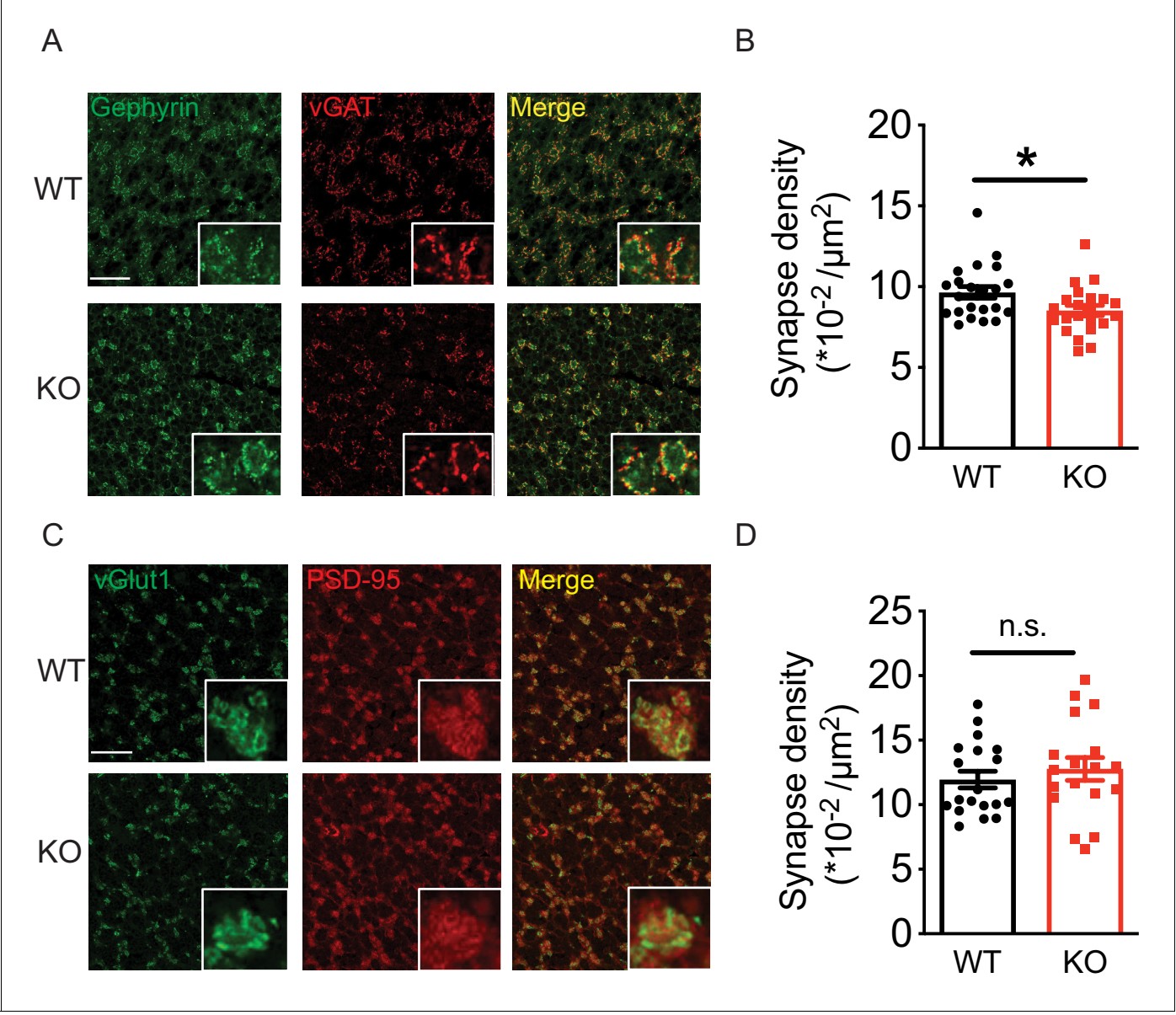

**Figure 7.** Loss of *Nxph4* reduced Golgi-granule inhibitory synapse number. (**A**) Gephyrin and vGAT staining in the cerebellar granular layer indicates Golgi-granule inhibitory synapses. (**B**) Quantification of puncta co-expressing gephyrin and vGAT as an indicator of inhibitory synapse number. (**C**) vGlut1 and PSD-95 staining in the cerebellar granular layer indicates mossy fiber-granule cell excitatory synapses. (**D**) Quantification of puncta co-expressing vGlut1 and PSD-95 as an indicator of excitatory synapse number. Data are presented as mean ± SEM. n.s., not significant; *p<0.05; by t test (**B**) or Mann-Whitney U test (**D**).

DOI: https://doi.org/10.7554/eLife.46773.017

The following figure supplement is available for figure 7:

**Figure supplement 1.** *Nxph4* KO mice showed normal expression and localization of GABA$_A$Rs.

DOI: https://doi.org/10.7554/eLife.46773.018

The selective expression of Nxph4 in only one type of neuron in a given brain region provides us the opportunity to study its role as an α-neurexin ligand in a synapse type-specific manner. Otherwise the specific function might be masked by the diverse roles neurexin plays at different types of neurons, including regulation of neurotransmitter release (*Missler et al., 2003*), synapse number (*Chen et al., 2017*), and postsynaptic receptors (*Aoto et al., 2013*). Various neurexin ligands likely mediate different synaptic functions, as neurexin forms different interaction networks with each of

the ligands. There are examples in excitatory synapses supporting this principle: in the cerebellar cortex, neurexin interacts with Cbln1 secreted by granule cells to control the matching and maintenance of parallel fiber (granule cell axons)-Purkinje cell synapses (*Elegheert et al., 2016*; *Hirai et al., 2005*; *Matsuda et al., 2010*; *Uemura et al., 2010*); in the CA3 region of the hippocampus, neurexin interacts with C1ql2/3 released from hippocampal mossy fibers to recruit postsynaptic kainate-type glutamate receptors (*Matsuda et al., 2016*). Here, we show that Nxph4, an α-neurexin cell type-specific synaptic partner, might facilitate the ability of neurexin to regulate inhibitory neurotransmission in the cerebellar Golgi-granule cell synapses. Therefore, the cell-specific ligands Nxph4, Cbln1, and C1ql2/3 modulate neurexin functions to facilitate its context-specific role. In addition, these three proteins are all secreted proteins, which may allow finer regulation of neurexin function in response to neuronal activity.

Our analysis of Golgi-granule cell connectivity using multi-cell patch recoding revealed decreased Golgi to granule neuron synaptic connectivity. We also detected strong reductions in mIPSC frequency and amplitude. The 50% reduction of mIPSC frequency in the KO mice, which is partially explained by the 15% less synapses in those mice, indicates impaired neurotransmitter release. Altogether, these findings support our conclusion that loss of Nxph4 reduced cerebellar Golgi-granule cell inhibitory synapses and impaired inhibitory neurotransmission. The synaptic phenotype displayed in the *Nxph4* KO mice resembled defects caused by impaired α-neurexin and GABA$_A$Rs. At the pre-synaptic sites, α-neurexin is required to couple Ca$^{2+}$ channels to the pre-synaptic machinery to trigger neurotransmitter release (*Missler et al., 2003*). Pan-α-neurexin KO reduced spontaneous neurotransmitter release in both the excitatory and inhibitory synapses in the neocortex and brainstem (*Missler et al., 2003*). Similar defects have also been observed when neurexin3 is ablated in inhibitory olfactory bulb neurons (*Aoto et al., 2015*), as well as when all α- and β-neurexins are deleted from the somatostatin-positive (SST$^+$) neurons in the prefrontal cortex (*Chen et al., 2017*). In terms of neurexin's role in regulating synapse formation and elimination, current data suggest that neurexins are not responsible for initiating synapse formation (*Missler et al., 2003*; *Südhof, 2018*). However, deletion of neurexins does cause loss of synapses in some specific cells such as parvalbumin-positive (Pv+) interneurons in the prefrontal cortex (*Chen et al., 2017*). Therefore, loss of Nxph4 might disrupt the Nxph4-α-neurexin complex and result in the reduced synapses and mIPSC frequency in the KO mice. Deletion of Nxph4 also impaired mIPSC amplitude, which might be due to loss of Nxph4 in the GABA$_A$Rs complex. It is possible that Nxph4 may mediate some of these synaptic functions through interacting with other ligands. These are issues that need to be addressed in future studies.

The fine regulation of Nxph4 on Golgi-granule cell inhibitory transmission is necessary to set the level of granule cell excitability in order to reconfigure all the converging inputs from mossy fibers (*D'Angelo and De Zeeuw, 2009*; *Galliano et al., 2010*). Without precise inhibition from the Golgi cells, the granule cells cannot generate an informative spatiotemporal map for the Purkinje cells, which are the sole cerebellar cortex output cells for controlling movement. Selectively ablating cerebellar Golgi cells in mice disrupts inhibition onto granule neurons and results in motor incoordination on the rotarod, highlighting the critical role of Golgi-granule cell inhibition in coordinating movement (*Watanabe et al., 1998*). Therefore, we propose that impaired inhibitory input onto granule cells due to loss of *Nxph4* might partially contribute to the motor deficits in the *Nxph4* KO mice. This, however, does not exclude the possible contribution of Nxph4 in other brain regions, especially the deep cerebellar nuclei. The vestibular nuclei may not be underlying the motor defects in the *Nxph4* KO mice as their function was not affected after loss of Nxph4 as illustrated by the proper righting reflex of the KO mice.

The present work suggests that Nxph4 has a critical role in regulating synapse functions in specific circuits possibly through interacting with α-neurexin and GABA$_A$ receptors. Synaptic complexes, such as the α-neurexin-Nxph4 complex we reported here, provide the basis for precise regulation of some of the diverse synapses. It is easy to imagine that altered expression of these proteins would lead to neuropsychiatric disorders, including autism spectrum disorders, schizophrenia, and depression (*Baudouin et al., 2012*; *Zoghbi and Bear, 2012*). This highlights a need for a more comprehensive understanding of synaptic biology and pathology in a cell type- and synapse type-specific level. We believe that deciphering how cell type-specifically expressed proteins regulate synapse function would give us a better understanding of how the brain controls behavior, which will make it possible

to achieve a unifying approach to treat these genetically heterogeneous but clinically overlapping neuropsychiatric disorders.

# Materials and methods

**Key resources table**

| Reagent type (species) or resource | Designation | Source or reference | Identifiers | Additional information |
|---|---|---|---|---|
| Antibody | Mouse monoclonal anti-FLAG M2 | Sigma-Aldrich | Cat# F1804 RRID:AB_262044 | (1:1000) |
| Antibody | Rabbit polyclonal anti-pan Neurexin-1 | Millipore | Cat# ABN161 RRID:AB_10917110 | (1:1000) |
| Antibody | Mouse monoclonal anti-HA1.1 | Bioledgend | Cat# 901513, RRID:AB_2565335 | (1:5000) |
| Antibody | Mouse monoclonal anti-Vinculin | Sigma-Aldrich | Cat# V9131, RRID:AB_477629 | (1:10000) |
| Antibody | Rabbit polyclonal anti-GABA$_A$R$\alpha$1 | Synaptic System | Cat# 224 203, RRID:AB_2232180 | (1:1000) |
| Antibody | Rabbit polyclonal anti-GABA$_A$R$\alpha$6 | Synaptic System | Cat# 224 603, RRID:AB_2619945 | (1:1000) |
| Antibody | Mouse monoclonal anti-Vglut1 | Millipore | Cat# MAB5502, RRID:AB_262185 | (1:1000) |
| Antibody | Guinea pig polyclonal anti-vGAT | Frontier Institute co. Ltd | Cat#: VGAT-GP-Af1000, RRID: AB_2571624 | (1:1000) |
| Antibody | Mouse monoclonal anti-beta Actin | Abcam | Cat#: ab20272 RRID:AB_445482 | (1:2000) |
| Antibody | Rabbit polyclonal anti-TGF beta 1 | Abcam | Cat#: ab92486, RRID:AB_10562492 | (1:1000) |
| Antibody | Rabbit polyclonal anti-MeCP2 | Zoghbi Lab | #0535 | (1:1000) |
| Antibody | Mouse monoclonal anti-Gephyrin | Synaptic system | Cat#: 14701, RRID:AB_887717 | (1:1000) |
| Antibody | Rabbit polyclonal anti-PSD-95 | Cell signaling | Cat # 2507, RRID:AB_561221 | (1:1000) |
| Recombinant DNA reagent | pLenti-mCherry | Addgene | Cat# 36084, RRID:Addgene_36084 | |
| Recombinant DNA reagent | pLenti-Nxph4-3xFLAG-mCherry | This paper | N/A | Results subsection 'Nxph4 is a secreted glycoprotein' |
| Recombinant DNA reagent | pLenti–Nxph4-4Q-HA-mCherry | This paper | N/A | Results subsection 'Nxph4 is a secreted glycoprotein' |
| Recombinant DNA reagent | pLenti–Nxph4-6A-HA-mCherry | This paper | N/A | Results subsection 'Nxph4 is a secreted glycoprotein' |
| Recombinant DNA reagent | pCMV-Nxph4-HA | This paper | N/A | Results subsection 'Nxph4 is a secreted glycoprotein' |
| Recombinant DNA reagent | pCMV-Nxph4-4Q-HA | This paper | N/A | Results subsection 'Nxph4 is a secreted glycoprotein' |
| Recombinant DNA reagent | pCMV-Nxph4-6A-HA | This paper | N/A | Results subsection 'Nxph4 interacts with $\alpha$-neurexin in vivo' |

*Continued on next page*

*Continued*

| Reagent type (species) or resource | Designation | Source or reference | Identifiers | Additional information |
|---|---|---|---|---|
| Recombinant DNA reagent | pCAG-HA-Nrxn1β S4(+) | Addgene | Cat# 59410, RRID:Addgene_59410 | |
| Recombinant DNA reagent | pCAG-HA-Nrxn1β S4(-) | Addgene | Cat# 59409, RRID:Addgene_59409 | |
| Recombinant DNA reagent | pCAG-HA-Nrxn1α | Addgene | Cat# 58266, RRID:Addgene_58266 | |
| Recombinant DNA reagent | pCAG-HA-Nrxn1α-LNS1 | This paper | N/A | Results subsection 'Nxph4 interacts with α-neurexin in vivo' |
| Recombinant DNA reagent | pCAG-HA-Nrxn1α-LNS2 | This paper | N/A | Results subsection 'Nxph4 interacts with α-neurexin in vivo' |
| Recombinant DNA reagent | pCAG-HA-Nrxn1α-LNS3 | This paper | N/A | Results subsection 'Nxph4 interacts with α-neurexin in vivo' |
| Recombinant DNA reagent | pCAG-HA-Nrxn1α-LNS4 | This paper | N/A | Results subsection 'Nxph4 interacts with α-neurexin in vivo' |
| Recombinant DNA reagent | pCAG-HA-Nrxn1α-LNS5 | This paper | N/A | Results subsection 'Nxph4 interacts with α-neurexin in vivo' |
| Recombinant DNA reagent | pCMV-GABARα1N-HA | This paper | N/A | Results subsection 'Nxph4 interacts with α-neurexin in vivo' |
| Recombinant DNA reagent | pCMV-GABARα6N-HA | This paper | N/A | Results subsection 'Nxph4 interacts with α-neurexin in vivo' |
| Chemical compound, drug | X-Gal | Thermo Fisher | Cat# 15520034 | |
| Chemical compound, drug | Lipofectamine 2000 | Thermo Fisher | Cat# 11668500 | |
| Commercial assay, kit | Antigen Unmasking Solution (citrate based) | Vector Laboratories | Cat# H-3300 | |
| Commercial assay, kit | Avidin/Biotin Blocking Kit | Vector Laboratories | Cat# SP-2001 | |
| Commercial assay, kit | TSA Kit | Invitrogen | Cat# T-20932 | |
| Commercial assay, kit | Papain Dissociation System Kit | Worthington Biochemical Corp. | Cat# LK003150 | |
| Commercial assay, kit | PNGase F | NEB | Cat# P0704S | |
| Commercial assay, kit | miRNeasy Mini Kit | Qiagen | Cat#: 217004 | |
| Commercial assay, kit | M-MLV Reverse Transcriptase | Thermo Fisher | Cat#: 28025013 | |
| Commercial assay, kit | iTaq Universal SYBR Green SuperMix | BIO-RAD | Cat#: 1725124 | |
| Cell line (*Homo sapiens*) | HEK293T cells | ATCC | RRID:CVCL_0063 | |
| Strain, strain background (*Mus musculus*) | Mouse: FVB | The Jackson Laboratory | Stock No: 001800 | |

*Continued on next page*

*Continued*

| Reagent type (species) or resource | Designation | Source or reference | Identifiers | Additional information |
|---|---|---|---|---|
| Strain, strain background (*Mus musculus*) | Mouse: $Nxph4^{\beta geo+/-}$ | This paper | N/A | The Jackson Laboratory: 033791 |
| Strain, strain background (*Mus musculus*) | Mouse: $Nxph4^{FLAG/FLAG}$ | This paper | N/A | The Jackson Laboratory: 033792 |
| Sequence-based reagent | $Nxph4^{\beta geo+/-}$ genotyping forward primer: 5'-AAAGACTAGCAGACGCAGCA | This paper | N/A | Materials and methods subsection '$Nxph4$-$^{\beta geo}$ mouse' |
| Sequence-based reagent | $Nxph4^{\beta geo+/-}$ genotyping reverse primer: 3'-CCCTAACTCCCCCAAACAGA | This paper | N/A | Materials and methods subsection '$Nxph4$-$^{\beta geo}$ mouse' |
| Sequence-based reagent | $Nxph4^{FLAG/FLAG}$ genotyping forward primer: 5'-TCAAGTTCTCGCTGTTGGTG | This paper | N/A | Materials and methods subsection '$Nxph4^{FLAG/FLAG}$ knock-in mouse' |
| Sequence-based reagent | $Nxph4^{FLAG/FLAG}$ genotyping reverse primer: 3'-TTCCACGTGGCAATTAAAAG | This paper | N/A | Materials and methods subsection '$Nxph4^{FLAG/FLAG}$ knock-in mouse' |

## Experimental model and subject details

### Mouse husbandry and handling

Mice were group housed in an AAALAS-certified animal facility on a 14 hr/10 hr light/dark cycle. All procedures to maintain and use these mice were approved by the Institutional Animal Care and Use committee for Baylor College of Medicine.

### *Nxph4-$\beta$geo* mouse

$Nxph4^{\beta geo+/-}$ mice were generated within our lab using standard methods of ESC injection into blastocysts followed by implantation into pseudopregnant female mice. Briefly, embryonic stem cells (strain background, C57BL/6N +Agouti mutation) possessing targeted insertion of a 'gene-trap' splice acceptor construct, containing an in-frame $\beta$-galactosidase cassette and a PGK-neomycin cassette, into intron 2 of the mouse endogenous Nxph4 locus on chromosome 10 were obtained from the Knock-out Mouse Project (KOMP) repository (*Austin et al., 2004*). This construct is designed to simultaneously disrupt endogenous Nxph4 transcript expression while driving expression of a $\beta$-galactosidase reporter gene in a precise promoter specific manner. Targeting in ESCs was confirmed by long-range PCR targeting the 5' and 3' junctions of the targeting construct. Chimeric mice were crossed to wild type C57BL/6J Albino mothers to detect germline transmission, and successful progenies were screened by PCR assay for presence of the *Nxph4-$\beta$geo* construct. Primers were 5'-AAA-GACTAGCAGACGCAGCA and 3'-CCCTAACTCCCCCAAACAGA. *Nxph4-$\beta$geo* positive mice were then expanded for further analysis by mating to wild type C57BL/6J mice. Mice at 14 to 16 weeks of age were used for most behavioral assays except for righting reflex with age stated in the results. Littermates of the same sex were randomly assigned to experimental groups.

### *Nxph4$^{FLAG/FLAG}$* knock-in mouse

The $Nxph4^{FLAG/FLAG}$ knock-in mouse was generated according to CRISPR genome editing method described previously (*Ran et al., 2013*). Briefly, an optimal sgRNA sequence (5'-gcggcacgatact-gacgctt) close to the genomic target site was chosen using the http://crispr.mit.edu/ design tool. The sgRNA was cloned into the pSpCas9(BB)−2A vector (pX330) via BbsI digestion and insertion site. T7 promoter was added to the sgRNA template by polymerase chain reaction (PCR). The PCR products were then purified and used as template for in vitro transcription using MEGAshortscript T7 Transcription Kit. sgRNA was purified with MEGAclear Transcription Clean-Up Kit. A dsDNA donor was designed with a linker sequence and 3xFLAG sequence inserted at the end of Nxph4

third domain (after 5'-ctccaagcgtgtggagttc). The PAM site (AGG) was mutated into AGA to avoid cleaving by Cas9. Two ~ 1.5 kb homologous arms on both ends flanking the inserted region were included into the dsDNA donor. At last, a 2945 kb dsDNA was synthesized for use as donor DNA (5'-agagaatcaacacagccacacaca...taatctaagcgtcagtatcgtgccgccc...gttcGGAGGCAGTGGGGGTAG TGGCGGGTCAGGAGGATCCGACTACAAGGACGACGATGACAAGGACTATAAGGACGATGATGA-CAAGGACTATAAGGATGACGATGACAAAGGCGGAAGTGGTGGCTCCGGGGGATCTGGGGGG TCAGGTGAGgggggcgtctg....gaagcggggacagcgtaggc, the inserted sequence is showed as upper case). To prepare protein mix for pronuclear injection, 4.5 μg Cas9 protein (Sigma) and 3 μg sgRNA were diluted into 150 μl buffer containing 10 mM Tris (pH = 7.5) and 0.25 mM EDTA, which was incubated in 37°C for 5 min. Then 4 μg donor dsDNA was added to the mix, followed by centrifugation at 20,000 xg for 10 min in 4°C. Top 100 μl was used for pronuclear injection to generate knock-in mice based on standard procedures. At last one founder mouse was identified with four different pairs of primers to confirm the correct insertion of the tag. Sanger sequencing was performed to confirm that intact donor DNA was inserted into the mouse genome. Knock-in mice were then crossed with wild type C57BL/6J mice for three generations before expending for further analysis. Positive progenies were identified by PCR assays with the following primers: 5'-TCAAGTTCTCGCTG TTGGTG and 3'-TTCCACGTGGCAATTAAAAG. Adult male and female mice were equally used in the experiments.

## HEK293T cell culture
HEK293T cells (authenticated by ATCC with STR profiling, mycoplasma contamination testing negative) were cultured in DMEM (Invitrogen) containing 10% FBS.

## Primary cortical neuron culture
Mouse cerebral cortices were removed from E16.5 embryos of wild type FVB mice and dissociated with papain dissociation kit (Worthington). The neurons were plated on poly-D-lysine-coated 12-well plates and maintained at 37°C incubator for ~13 days. Neurons were treated with lentivirus at DIV 1.

## Method details
### β-galactosidase staining
β-galactosidase staining was performed essentially according to *Juntti et al. (2010)*. Both male and female *Nxph4*$^{βgeo+/-}$ mice were used and did not display obvious difference in β-galactosidase staining. Mice for β-galactosidase staining were transcardially perfused with 1xPBS followed by 4% paraformaldehyde. Brains were dissected out and post-fixed in 4% paraformaldehyde for another 2 hr followed by dehydrating in 20% sucrose solution overnight. Brains were then embedded in optimum cutting temperature (O.C.T.; Sakura Finetak) and frozen at −80°C. Sectioning was performed on a Leica cryostat at 25 μM per section. Mounted sections were washed with 3 changes of solution A (1X PBS, 2 mM MgCl$_2$) for 5 min each. Slides were then incubated in pre-warmed solution B (0.1 M NaPO$_4$, 2 mM MgCl$_2$, 0.02% NP40, 0.01% sodium deoxycholate) for 10 min followed by incubation in X-gal working solution (0.6 mg/ml X-gal, 0.5 M potassium ferrocyanide, 0.5 M potassium ferricyanide in solution B) at 37°C for ~36 hr. Sections were then washed in 1X PBS and then counter-stained with Nuclear Fast Red. Slides were mounted in aqueous mounting medium (Aqua Permount) and cover-slipped.

## RNA In situ hybridization
RNA in situ hybridization (ISH) was performed on 25 μm thick coronal sections cut from fresh frozen adult WT/*Nxph4* KO mice. We generated digoxigenin (DIG)-labeled mRNA antisense probes against Nxph4 and a fluorescein (FITC)-labeled probe against Gad1, vGlut1 and vGlut2 using reverse-transcribed mouse cDNA as a template. Both DIG- and FITC-labeled probes were made using RNA labeling kits from Roche. Primer and probe sequences for the Nxph4, vGlut1, and vGlut2 probes were based on the published sequences in Allen Brain Atlas and the Gad1 was based on the probe sequence described in EurExpress.

ISH was performed by the RNA In Situ Hybridization Core at Baylor College of Medicine using an automated robotic platform as previously described (*Yaylaoglu et al., 2005*), with modifications of the protocol for double ISH. Modifications in brief (see *Yaylaoglu et al., 2005*) for buffer

descriptions): both probes were hybridized to the tissue simultaneously (Nxph4/Gad1, Nxph4/vGlut1 or Nxph4/vGlut2). After the described washes and blocking steps the DIG-labeled probes were visualized using tyramide-Cy3 Plus (1/50 dilution, 15 min incubation, Perkin Elmer). After washes in TNT the remaining HRP-activity was quenched by a 10 min incubation in 0.2 M HCl. The sections were then washed in TNT, blocked in TNB for 15 min before a 30 min room temperature incubation with HRP-labeled sheep anti-FITC antibody (1/500 in TNB, Roche). After washes in TNT the FITC-labeled probe was visualized using tyramide-FITC Plus (1/50 dilution, 15 min incubation, Perkin Elmer). Following washes in TNT the slides were removed from the machine and mounted in ProLong Diamond with DAPI (Molecular Probes).

## RT-qPCR

Cerebella from adult WT and *Nxph4* KO mice were dissected and processed by miRNeasy mini kit to collect RNA (Qiagen, 74104). First-strand cDNA was synthesized using M-MLV reverse transcriptase (Life Technologies). We performed qPCR with Bio-Rad CFX96 Real-Time system using iTaq Universal SYBR Green SuperMix (BIO-RAD). The relative amount of cDNA was determined based on the cycle threshold. All reactions were conducted in triplicate and the results were averaged for each sample, normalized to *Ppia* levels, and analyzed using the ddCt method. Relative expression level of *Nxph4* was determined by normalizing the expression level of each sample to the average of WT controls. The following primers were used in the experiment:

> *Nxph4*: Forward 5'-GTGAGCACCCCTACTTTGGA-3', Reverse 5'-AAGGCTGTTTTTCTCCACCA-3'
> *Ppia*: Forward 5'-GCATACAGGTCCTGGCATCT-3', Reverse 5'-CCATCCAGCCATTCAGTCTT-3'.

## Behavioral assays

All the behavioral assays were carried out blinded to the genotype. Mice at 14 to 16 weeks of age were habituated in the testing room for at least 30 min before the test.

## Accelerating rotarod

Mice were placed on an accelerating rotarod (Ugo Basile) whose speed increased from 4 to 40 rpm over a five-minute period. Each animal was tested in four trials per day for four consecutive days, with a 30 min interval between two trials in the same day. Latency to fall was recorded when the mouse fell from the rod or when the mouse had ridden the rotating rod for two revolutions without regaining control. Data are shown as mean ± standard error of mean and analyzed by two-way ANOVA with Tukey's post hoc analysis.

## Open field assay

After habituation in the testing room (200 lux, 60 dB white noise), mice were individually placed in the center of an open Plexiglas chamber (40 × 40 × 30 cm) with photo beams (Accuscan) to measure their activity for 30 min. Data are shown as mean ± standard error of mean and analyzed by one-way ANOVA with Tukey's post hoc analysis.

## Elevated plus maze

After habituation to the testing room (200 lux, 60 dB white noise), the mouse was placed in the center of a four-arm maze (each arm 25 × 7.5 cm), with two opposing arms enclosed by 15 cm high walls and the other two open. The maze was 50 cm above the ground level. Activity was recorded by a suspended digital camera and ANY-maze (Stoelting Co.) video tracking software for 10 min. Data are shown as mean ± standard error of mean and analyzed by one-way ANOVA with Tukey's post hoc analysis.

## Acoustic startle response and Pre-pulse inhibition

Mice were habituated outside the test room for 30 min. Each mouse was placed in a Plexiglas tube inside of a sound-insulated lighted box (SR-Lab, San Diego Instruments). Startle stimulus was 120 dB and three pre-pulses used were 74, 78, and 82 dB. Pre-pulse inhibition was calculated as 1-[averaged startle response to startle stimulus with pre-pulse/averaged response to startle stimulus] x 100. Data are shown as mean ± standard error of mean. Acoustic startle response data are analyzed by one-

way ANOVA with Tukey's post hoc analysis. Pre-pulse Inhibition data are analyzed by two-way ANOVA with Tukey's post hoc analysis.

## Righting reflex

At postnatal day 10, pups were put on their back for 5 s on the bench and then were released. The time that pups spent to return to prone position was recorded. We performed this test before knowing the genotype of the pups, therefore more HET than WT and KO mice were tested. Data are analyzed by one-way ANOVA.

## Lentivirus production

Lentivirus was produced by transfecting HEK293T cells with lentiviral transfer plasmids, packaging plasmids, and envelope plasmids using Lipofectamine 2000 reagent (Invitrogen). Media from transfected cells were harvested 48 and 72 hr after transfection. Viruses were concentrated from the media using Lenti-X concentrator (Clontech) for further use.

## Protein analysis for HEK293T cells

HEK293T cells (obtained and certified from ATCC) were cultured in DMEM (Invitrogen) containing 10% FBS. After transfection (Lipofectamine 2000, Invitrogen), cells were cultured for another 48 hr and then lysed with 2% TX-100 buffer (150 mM NaCl, 50 mM Tris-HCl [pH 7.5], 2% TX-100, protease and phosphatase inhibitors (Gendepot)). After 10 min incubation on ice, lysates were centrifuged at 17,000 g, 10 min, 4°C, and supernatant was then used for immunoblotting or IP. For immunoblotting, sample buffer and reducing agent were mixed with each sample followed by a 10 min incubation at 70°C. Samples were then run on a 4–12% Bis-Tris gel, transferred to a PVDF membrane and blocked for one hour with 5% non-fat milk prior to primary antibody incubation. For IP, 20 µl antibody conjugated beads (anti-FLAG magnetic beads, Sigma-Aldrich, M8823 or anti-HA magnetic beads, Fisher, 88836) were added to the sample followed by overnight incubation at 4°C with rotation. Beads were then washed with 3 × 1000 µl of 0.2% TX-100 buffer (150 mM NaCl, 50 mM Tris-HCl [pH 7.5], 0.2% TX-100, protease inhibitors (Gendepot), and phosphatase inhibitor (Gendepot)) before being eluted in 2X elution buffer at 95°C for ten minutes.

## Protein analysis for mouse brains

To purify synaptosomes to prove the presence of Nxph4 on synapses, mouse cerebella was homogenized with 20 strokes by TKA EUROSTAR 900 rpm in sucrose homogenization buffer (320 mM sucrose, 10 mM Tric-HCl [pH 7.5], 5 mM EDTA, protease and phosphatase inhibitors (Gendepot)). The lysates (H) were then centrifuged at 900 g for 10 min at 4°C. The supernatant (S1) was further centrifuged at 17,000 g for 20 min at 4°C. Supernatant (cytosolic fraction, S2) was removed and the pellet (crude synaptosomes, P2) was resuspended in sucrose homogenization buffer followed by sucrose gradient ultracentrifugation at 64,000 g for 2 hr (0.8/1.0/1.2 M sucrose gradients). The interface of 1.0/1.2 M was collected and centrifuged at 164,000 g for 1 hr. The resulting pellet, the synaptosomes (Syn), was dissolved in sucrose homogenization buffer for analysis.

The crude synaptosomes were used for IP experiments. To pull down Nxph4-3xFLAG, mouse brains from two KI mice or two wild type mice containing olfactory bulb, hypothalamus, midbrain, hindbrain, and the cerebellum were used. Crude synaptosomes were resuspended in 1 ml TE buffer (10 mM Tris-HCl [pH 7.5], 5 mM EDTA, protease and phosphatase inhibitors (Gendepot)) and 120 µl DOC buffer (10% sodium deoxycholate, 500 mM Tris-HCl [pH 9.0]) followed by a 30 min incubation in 36°C with shaking. After the incubation, 130 µl buffer T (1% Triton X-100, 500 mM Tris-HCl, [pH9.0]) was added to the sample, which was then subjected to dialysis against 1 L binding/dialysis buffer (50 mM Tris-HCl [pH7.5], 0.1% Triton X-100) in a dialysis tubing (Pierce, HK108503) with rotation overnight at 4°C. On the next day, samples were centrifuged at 17000 g for 40 min at 4°C. The supernatant was collected and concentration was determined by BCA assay. Two percent of the sample was saved as input. Equal amount of total protein from the KI and wild type mice were mixed with 20 µl anti-FLAG magnetic beads separately, followed by overnight incubation at 4°C with rotation. Beads were then washed with 3 × 1000 µl of binding/dialysis buffer before being eluted in 2X elution buffer at 95°C for ten minutes.

To detect Nxph4-GABA$_A$Rs complex, mouse cerebella were processed similarly as above to get crude synaptosomes, which was then resuspended in 1% TX-100 buffer (150 mM NaCl, 50 mM Tris-HCl [pH 7.5], 1% TX-100, protease and phosphatase inhibitors (Gendepot)) followed by a 30 min incubation on ice. The supernatant was collected and concentration was determined by BCA assay. Two percent of the sample was saved as input. Normal rabbit IgG (Millipore, 12370) or 5 ul anti-α-neurexin antibody (Millipore, ABN161; ABN161-I) was added to the sample followed by overnight incubation at 4°C with rotation. The next day, 20 μl Protein A Sepharose (Fisher, 17-0780-01) was added followed by a one-hour incubation. Beads were then washed with 3 × 1000 μl of 0.2% TX-100 buffer (150 mM NaCl, 50 mM Tris-HCl [pH 7.5], 0.2% TX-100, protease inhibitor and phosphatase inhibitors (Gendepot)) before being eluted in 2X elution buffer at 95°C for ten minutes.

## Deglycosylation

PNGase F (NEB, P0704S) was used based on the manufacture manual. After being mixed with glyco-protein denaturing buffer, cell or brain lysates were denatured by heating at 100°C for 10 min. The denatured protein was then mixed with G7 reaction buffer, NP40, and PNGaseF. The sample was incubated at 37°C for 1 hr for deglycosylation.

## Cerebellar slice electrophysiology

Mice (sIPSCs, and eEPSCs:~3 months old; *Nxph4* KO and WT mIPSCs:~P25-35; *Nxph4* KI and WT mIPSCs:~P30-40) were anesthetized by isoflurane inhalation and decapitated immediately. Acute cerebellar sagittal slices (350 μm thick) were cut with a vibratome (LEICA VT 1200, Leica Microsystems Ins., Buffalo Grove, IL) in a chamber filled with cutting solution. The sucrose containing cutting solution was used to prepare slices for mIPSC recording (*Egawa et al., 2012*; *Pan et al., 2009*) including (in mM) 235 sucrose, 2.5 KCl, 1.25 NaH$_2$PO$_4$, 28 NaHCO$_3$, 0.5 CaCl$_2$, 7 MgSO$_4$, 28 D-glucose. The choline-chloride cutting solution was used to prepare slices for sIPSC recording, which included (in mM) 110 choline-chloride, 25 NaHCO$_3$, 25 D-glucose, 11.6 sodium ascorbate, 7 MgSO$_4$, 3.1 sodium pyruvate, 2.5 KCl, 1.25 NaH$_2$PO$_4$, and 0.5 CaCl$_2$ with atropine (20 μM). The slices were then incubated in artificial cerebrospinal fluid (ACSF, in mM) containing 126 NaCl, 26 NaHCO$_3$, 20 D-glucose, 2.5 KCl, 2.0 CaCl$_2$, 2.0 MgSO$_4$, and 1.25 NaH$_2$PO$_4$ at room temperature after recovery at 37 ± 1°C for 30 min. The solutions were bubbled through with 95% O$_2$ and 5% CO$_2$.

Whole-cell recordings were performed using a patch clamp amplifier (MultiClamp 700B; Molecular Devices, Sunnyvale, CA). Data acquisition and analysis were performed using a digitizer (DigiData 1440) and software (Minianalysis 6.0.3, Synaptosoft Inc). Signals were filtered at 2 kHz and sampled at 10 kHz. Microelectrodes with resistance of 5–7 MΩ were pulled from borosilicate glass capillaries (Sutter Instruments, Novato, CA). The intra-pipette solution to measure sIPSC contained (in mM) 120 CsCH$_3$SO$_3$, 20 HEPES, 0.4 EGTA, 5 TEA-Cl (tetraethylammonium chloride), 2 MgCl$_2$, 2.5 MgATP, 0.3 GTP, 10 Na$_2$-phosphocreatine, and 1 QX-314 [N-(2,6-dimethylphenylcarbamoylmethyl) triethylammonium bromide] (pH 7.2 with CsOH). The granular cells were voltage-clamped at +10 mV for sIPSC. The experiment was performed at 30 ± 1°C using an automatic temperature controller (Warner Instrument, Hamden, CT). Miniature IPSCs were recorded at a holding potential of −70 mV in the presence of (in μM) 10 6-cyano-7-nitroquinoxaline-2, 3-dione (CNQX), 50 D-2-amino-5-phos-phonopentanoic acid (AP5), 0.3 strychnine hydrochloride, and 1.0 tetrodotoxin (TTX). The high-Cl-intra-pipette solution contained (in mM): 140 CsCl, 9 NaCl, 1 MgCl$_2$, 1 EGTA, and 10 HEPES (pH 7.3, adjusted with KOH). ACSF with 18 mM [K$^+$] was used to increase mIPSC frequency during electrophysiological recordings (*Accardi et al., 2015*; *Momiyama and Takahashi, 1994*). Accordingly, sucrose was added to internal solution to maintain osmolarity at 330 mOsmol/L. The intra-pipette solution to record eEPSC contained (in mM) 140 Cs-gluconate, 15 HEPES, 0.5 EGTA, 2 TEA-Cl, 2 MgATP, 0.3 Na$_3$GTP, 10 Na$_2$-phosphocreatine and 2 QX314-Cl (pH was adjusted to 7.2 with CsOH). Mossy fibers were stimulated with a bipolar tungsten electrode at an interval of 15 s via ISO-Flex unit and Master-8 (A.M.P.I. Israel) in the presence of picrotoxin (50 μM). The experiments were performed at 25 ± 1°C.

Simultaneous multi-cell patch recordings were obtained from neurons in granular layers of cerebellar slices as described previously (*Jiang et al., 2015*; *Jiang et al., 2013*). P21-P31 male mice were used in the experiment. Briefly, patch pipettes (5–7 MΩ) were filled with intracellular solution containing 121 mM potassium gluconate, 10 mM HEPES, 25 mM KCl, 4 mM MgATP, 0.3 mM Na$_3$GTP,

and 10 mM sodium phosphocreatine (pH 7.25). Whole-cell recordings were made from up to eight neurons with two Quadro EPC 10 amplifiers (HEKA Electronics, Lambrecht, Germany). A built-in LIH 8+8 interface board (HEKA) was used to achieve simultaneous A/D and D/A conversion of current, voltage, command and triggering signal for up to eight amplifiers. Patch Master software (HEKA) and custom-written Matlab-based programs (Mathworks) were used to operate the recording system and perform online and offline analysis of the electrophysiology data (*Hao, 2019*; copy archived at https://github.com/elifesciences-publications/PatchClamp-ShowConnection) (*Jiang et al., 2015*; *Jiang et al., 2013*). Action potentials (APs) were evoked by current injection into the presynaptic Golgi cells at 2 nA for 2 ms at 0.1 Hz for 30–50 trials, and the average of the sweeps in postsynaptic granule cells was used to detect synaptic connections (connection or no connection), and calculate the basic properties of evoked unitary postsynaptic potentials (uPSPs), such as synaptic latency, 10–90% rise time and decay time constant if two cells were connected. The recordings were performed in the presence of AMPA receptor antagonist CNQX (20 µM) and NMDA receptor antagonist DL-APV (100 µM) to only detect GABAergic synaptic transmission. We considered the synaptic response as a postsynaptic event when the amplitude of the response was larger than 2 times the baseline standard deviation. In addition to detecting the synaptic connection between cell pair, we also recorded the firing pattern and intrinsic electrophysiological properties of all recorded neurons.

## Immunostaining

Perfused mouse brains were incubated in 4% PFA at 4°C overnight and protected by 30% sucrose before embedding in O.C.T. Frozen sections were cut with 20 or 45 µm thickness, and O.C.T. was removed by incubating in 1X PBS at RT for 30 min.

For FLAG staining, 20 µm sections were permeablized with 1X PBS containing 0.1% TritonX-100 (PBST) at RT for 30 min. To quench endogenous oxidase, slides were treated with 2% hydrogen peroxide in 1X PBS at RT for 30 min. Antigen retrieval was performed by incubating slides in boiled Antigen Unmasking Solution, Citric Acid Based (Vector Laboratories) for 20 min and chilled at RT for another 20 min. Samples were blocked with 10% normal donkey serum in PBST containing avidin (Avidin/Biotin Blocking Kit; Vector Laboratories) at RT for at least 2 hr. Sections were incubated with an anti-FLAG antibody in PBST containing biotin (Avidin/Biotin Blocking Kit; Vector Laboratories) at 4°C overnight. Secondary donkey anti-rabbit antibody was conjugated with biotin (Jackson ImmunoResearch Labs) with 1:1000 dilution in PBST. Samples were then incubated with the secondary antibody mix at RT for 2 hr. The signal was then developed by incubating the sections in horseradish peroxidase (HRP)-conjugated streptavidin (1:100) and labeled tyramide (1:100) according to the manufacturer manual (TSA Kit, Invitrogen). We used a Leica TCS SP8 confocal system to detect fluorescent staining.

For synaptic markers and GABARs staining, perfused mouse brains were embedded in paraffin and cut with 6 µm thickness. Sections were then rehydrated with xylene followed by antigen retrieval with citric acid. Samples were stained overnight with primary antibody, 2 hr with secondary antibody, and mounted with Vectashield hardset antifade mounting medium. Images of the granular layer were acquired with 63x or 100x magnification using a Leica TCS SP8 confocal system. To count synapse number, images were analyzed using Imaris spot analysis tool. Briefly, with a given detection threshold, which was the same for each marker across samples, the software counted the number of puncta expressing each marker. It also counted among those puncta how many were very close to pucncta expressing another marker, which was considered as co-localization/co-expression. GABARs clusters density and size were analyzed using Fiji software. First, a region in the granular layer was set as a region of interest. Then we used the built-in rolling ball method to subtract background. We further used the watershed function to better separate puncta. The number and area of puncta were measured by the Analyze Particles function in Fiji. The density was calculated by the number of puncta divided by the area of ROI. The same settings were used for all the samples. Data were analyzed by an individual blinded to the genotype.

## Quantification and statistical analysis

Data were analyzed using student's t test, Mann-Whitney U test, one-way or two-way ANOVA followed by Tukey's post hoc analysis in GraphPad Prism 6 and 8 or SPSS. A p-value cut-off of 0.05 was considered statistically significant. Sample sizes were determined by previous experience

(*Chao et al., 2010*; *Meng et al., 2016*). Number of animals and cells used for each experiment can be found in the figure legends. Data were presented as mean ± standard error of the mean (SEM).

## Acknowledgements

We thank Vicky Brandt for critical review of the manuscript. This work was supported by NIH/NINDS grant 5R01NS057819 (HYZ) and partly by the Microscopy Core, Neurobehavior Core, Neuropathology Core, and RNA In Situ Hybridization Core of IDDRC at Baylor College of Medicine (1U54HD083092) from the Eunice Kennedy Shriver National Institute of Child Health and Human Development. The content is solely the responsibility of the authors and does not necessarily represent the official views of the Eunice Kennedy Shriver National Institute of Child Health and Human Development or the National Institutes of Health. We thank the Mouse ES Cell Core and the Genetically Engineered Mouse Core that are partially supported by the National Institutes of Health (NIH) grant P30CA125123 at Baylor College of Medicine for generating *Nxph4-3xFLAG* knock-in mouse models. HYZ is an investigator with the Howard Hughes Medical Institute.

## Additional information

### Competing interests

Huda Y Zoghbi: Senior editor, *eLife*. The other authors declare that no competing interests exist.

### Funding

| Funder | Grant reference number | Author |
|---|---|---|
| National Institute of Neurological Disorders and Stroke | 5R01NS057819 | Huda Y Zoghbi |
| Autism Speaks | 9120 | Li Wang |
| National Institute of Mental Health | R01MH117089 | Mingshan Xue |
| Brain and Behavior Research Foundation | NARSAD Young Investigator Award | Mingshan Xue |
| NIH Office of the Director | F31NS101891 | Amanda M Brown |
| American Epilepsy Society | Postdoctoral fellowship | Wu Chen |
| National Institute of Neurological Disorders and Stroke | R01NS100893 | Mingshan Xue |
| National Institute of Neurological Disorders and Stroke | R01NS100874 | Roy V Sillitoe |
| National Institute of Neurological Disorders and Stroke | R01NS089664 | Roy V Sillitoe |

The funders had no role in study design, data collection and interpretation, or the decision to submit the work for publication.

### Author contributions

Xiangling Meng, Conceptualization, Resources, Data curation, Formal analysis, Validation, Investigation, Visualization, Methodology, Writing—original draft, Project administration, Writing—review and editing; Christopher M McGraw, Conceptualization, Resources, Formal analysis, Investigation, Methodology, Writing—review and editing; Wei Wang, Formal analysis, Investigation, Methodology, Writing—review and editing; Junzhan Jing, Data curation, Formal analysis, Methodology; Szu-Ying Yeh, Amanda M Brown, Investigation, Methodology, Writing—review and editing; Li Wang, Funding acquisition, Investigation, Writing—review and editing; Joanna Lopez, Investigation, Writing—review and editing; Tao Lin, Validation, Investigation, Writing—review and editing; Wu Chen, Software, Investigation, Writing—review and editing; Mingshan Xue, Roy V Sillitoe, Supervision, Funding acquisition, Writing—review and editing; Xiaolong Jiang, Formal

analysis, Supervision, Methodology; Huda Y Zoghbi, Conceptualization, Supervision, Funding acquisition, Project administration, Writing—review and editing

### Author ORCIDs
Xiangling Meng https://orcid.org/0000-0002-3528-4037
Szu-Ying Yeh http://orcid.org/0000-0003-4506-5652
Mingshan Xue https://orcid.org/0000-0003-1463-8884
Roy V Sillitoe https://orcid.org/0000-0002-6177-6190
Huda Y Zoghbi https://orcid.org/0000-0002-0700-3349

### Ethics
Animal experimentation: Mice were housed in an AAALAS-certified animal facility. All procedures to maintain and use these mice were approved by the Institutional Animal Care and Use committee for Baylor College of Medicine. Animal protocol number AN-1013.

### Decision letter and Author response
Decision letter https://doi.org/10.7554/eLife.46773.021
Author response https://doi.org/10.7554/eLife.46773.022

## Additional files

### Supplementary files
• Transparent reporting form
DOI: https://doi.org/10.7554/eLife.46773.019

### Data availability
All data generated or analysed during this study are included in the manuscript and supporting files. Custom-written Matlab-based programs used to operate the recording system and perform online and offline analysis of the electrophysiology data have been made available at https://github.com/haozhaozhe/PatchClamp-ShowConnection (copy archived at https://github.com/elifesciences-publications/PatchClamp-ShowConnection).

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
