## [Decision Letter]

Thank you for submitting your article "Nxph4 is a selectively expressed α-neurexin ligand that modulates specific cerebellar synapses and motor functions" for consideration by *eLife*. Your article has been reviewed by three peer reviewers, one of whom is a member of our Board of Reviewing Editors, and the evaluation has been overseen by Catherine Dulac as the Senior Editor. The following individual involved in review of your submission has agreed to reveal their identity: Mary E Hatten (Reviewer #3).

The reviewers have discussed the reviews with one another and the Reviewing Editor has drafted this decision to help you prepare a revised submission.

Summary:

Neurexophilin4 is a soluble neuropeptide-like molecule with poorly understood role in synapse organization. In the current study, the authors generate a mouse model that enables investigation of neurexophilin4 localization as well as its conditional deletion. Anatomical analysis indicates that neurexophilin4 is expressed in various brain regions controlling appetitive behaviors and fluid intake. Functional analysis focuses on the Golgi cells within the cerebellum and show a significant reduction in their inhibitory inputs onto granule cells. The authors also show that neurexophilin4 (similar neurexophilin1-3) is a secreted glycol protein that selectively binds to α-neurexins. These data provide important new insights into the functions of Nxph4 protein in the mouse brain. The manuscript is very well-written, the data in large part are convincing and the experiments appear well conducted. However, there are a few important points that need to be addressed.

Essential revisions:

1) While molecular and anatomical analysis is extensive, electrophysiological analysis is rather limited in scope. Given the interesting phenotype associated with mIPSCs onto granule cells, it is critical to strengthen this finding by examining evoked inhibitory neurotransmission as well as excitatory neurotransmission under the same conditions.

2) Although the authors do not detect a change in GABA receptor expression levels, a potential alteration in their synaptic clustering could account for the decrease in mIPSC amplitudes. The manuscript would significantly benefit from more quantitative image analysis of GABAR cluster size and density.

3) Figure 2, the Nxph4 KO mice display pretty severe motor coordination deficits. Therefore, it is important to know whether the reduced anxiety phenotype observed in the KOs is a secondary effect of low motor function rather than a bonafide mood-related phenotype. It seems like open field observations were conducted in these mice (Materials and methods). Careful analyses of that data may provide answers towards this point.

4) Figure 2C and D, each animal should be shown as a point (rather than just a bar graph) to see the distribution.

5) Figure 6C, D, E, and F; each cell recorded should be shown as a data point so the reader can see their distribution. Did the data pass normality check? What statistical methods used to analyze them post-test, please clarify?

6) It is important to provide cumulative probability plots for Figure 6C and D and run the correct stats on those plots as well.

7) In Figure 6E and F, examples of waveforms should be included.

8) To quantify the number of structural synapses by immunohistochemistry a combination of pre and postsynaptic markers should be utilized and the apparent co-localization (due to the extremely close proximity of the pre and postsynaptic specializations at the synapse) should be calculated. The number or size of presynaptic markers may change without a real change in synapse numbers, or the synapse numbers may change but not the puncta appearance of presynaptic markers. It is also important to use correct synaptic pairs for these analyses. Sometimes in the literature, you may see the use of a presynaptic marker (such as VGlut1) colocalization with dendritic marker MAP2. This is incorrect and will yield co-localization driven by chance rather than a specific synaptic localization. VGAT and gephyrin would work for inhibitory synapses, and VGlut1 and PSD95 would work for parallel fiber excitatory synapses. Staining, imaging, and analyses should be done blind to genotype as well as the analyses.

---

## [Author Response]

Essential revisions:1) While molecular and anatomical analysis is extensive, electrophysiological analysis is rather limited in scope. Given the interesting phenotype associated with mIPSCs onto granule cells, it is critical to strengthen this finding by examining evoked inhibitory neurotransmission as well as excitatory neurotransmission under the same conditions.

We extended the electrophysiological studies by examining granule cell spontaneous IPSC, which revealed significantly impaired frequency in the KO mice (Figure 6B-D). This, together with the reduced mIPSC, further confirmed the diminished inhibition onto granule cells due to deletion of Nxph4. To directly test if loss of Nxph4 impaired Golgi-granule inhibitory synapses and thus resulted in the reduced inhibitory synaptic events detected in KO mice, we performed multi-cell recording to investigate the connectivity rate of Golgi-granule synapses, which turned out to be lower in the KO mice (Figure 6H-J). These data provide direct evidence supporting impaired Golgi-granule inhibitory synapses.

To examine excitatory neurotransmission, we planned to record both mEPSC and evoked EPSC on granule cells. However, the frequency of granule cell spontaneous excitatory synaptic activity was very low. It would then need a larger than usual sample size and a longer recording duration to characterize mEPSC. Meanwhile, with the paired-pulse stimulation protocol to evoke EPSC, we can measure properties of both pre and postsynaptic sides. Thus, evoked EPSCs are a better way to investigate neurotransmission of mossy fibers-granule cell excitatory synapses. We found out that the evoked EPSCs were similar in WT and KO mice (Figure 6—figure supplement 2), suggesting that deletion of Nxph4 did not significantly affect excitatory synapse function.

2) Although the authors do not detect a change in GABA receptor expression levels, a potential alteration in their synaptic clustering could account for the decrease in mIPSC amplitudes. The manuscript would significantly benefit from more quantitative image analysis of GABAR cluster size and density.

We agree with the reviewers and performed GABAAR immunofluorescence staining in another batch of samples. To better analyze GABAAR clusters, we captured images with 100x objective lens. We analyzed cluster size and density with Fiji software but did not detect significant differences between genotype (Figure 7—figure supplement 1C and D).

3) Figure 2, the Nxph4 KO mice display pretty severe motor coordination deficits. Therefore, it is important to know whether the reduced anxiety phenotype observed in the KOs is a secondary effect of low motor function rather than a bonafide mood-related phenotype. It seems like open field observations were conducted in these mice (Materials and methods). Careful analyses of that data may provide answers towards this point.

We performed open field test on *Nxph4* KO and WT mice and presented these data in Figure 2—figure supplement 1E and F. *Nxph4* KO mice travelled similar distance as WT mice, indicating normal locomotor function. Therefore, the longer time KO mice spent in open arms of the elevated plus maze should reflect their anxiety level instead of motor function.

4) Figure 2C and D, each animal should be shown as a point (rather than just a bar graph) to see the distribution.

As suggested, we now showed data points representing each animal in the revised paper in Figure 2C and D.

5) Figure 6C, D, E, and F; each cell recorded should be shown as a data point so the reader can see their distribution. Did the data pass normality check? What statistical methods used to analyze them post-test, please clarify?

In the revised paper we presented each data point to show the distribution for the figures mentioned above (Figure 6F and G, Figure 6—figure supplement 1). mIPSC amplitude and decay time passed normality check, we then analyzed them by t-test. mIPSC frequency and rise time did not show normal distribution, we then conducted Mann Whitney U test. In the revised paper we clarified this part in the figure legends.

6) It is important to provide cumulative probability plots for Figure 6C and D and run the correct stats on those plots as well.

We now show the cumulative probability plots in Figure 6F and G and conducted Kolmogorov- Smirnov test, which revealed significant differences between WT and KO.

7) In Figure 6E and F, examples of waveforms should be included.

We now show mIPSC waveforms in Figure 6E.

8) To quantify the number of structural synapses by immunohistochemistry a combination of pre and postsynaptic markers should be utilized and the apparent co-localization (due to the extremely close proximity of the pre and postsynaptic specializations at the synapse) should be calculated. The number or size of presynaptic markers may change without a real change in synapse numbers, or the synapse numbers may change but not the puncta appearance of presynaptic markers. It is also important to use correct synaptic pairs for these analyses. Sometimes in the literature, you may see the use of a presynaptic marker (such as VGlut1) colocalization with dendritic marker MAP2. This is incorrect and will yield co-localization driven by chance rather than a specific synaptic localization. VGAT and gephyrin would work for inhibitory synapses, and VGlut1 and PSD95 would work for parallel fiber excitatory synapses. Staining, imaging, and analyses should be done blind to genotype as well as the analyses.

We improved the quantification of synapse number by using combinations of synaptic markers as the reviewer suggested. VGlut1 and PSD95 were used for excitatory synapses, while VGAT and gephyrin were used for inhibitory synapses. We counted the number of puncta expressing both the pre and postsynaptic markers. Consistent with our previous results, both genotypes have similar mossy fibers-granule cell excitatory synapses (Figure 7C and D). However, the KO mice showed significantly reduced inhibitory synapse number in the granular layer compared with WT (Figure 7A and B), suggesting that loss of Nxph4 impaired cerebellar Golgi-granule inhibitory synapse number. Staining, imaging, and analyses were performed with the researchers being blind to genotype.

PSD-95 staining in Figure 7C does not show perfect puncta-like structure, which is most likely due to low and clustered PSD-95 expression in this region. In the cerebellar molecular layers of the same brain section we saw better puncta like PSD-95 staining, suggesting that both the anti- PSD-95 antibody and our experimental techniques are good. The current images worked well for quantification.